# A Hybrid Active Learning Regression Approach for Accelerating Annotation with Data Generation Constraints

## Abstract

In numerous scientific scenarios, experimental samples are designed as multiple data groups based on their underlying structures, *e.g.,* with 1000 samples in each group, where these samples share certain similarities but include systematic physicochemical variations. Then, a smaller number of samples (*e.g.,* 10) is selected to be placed in the parallel synthesizer, under a lengthy process, to conduct synthesis and collect their properties for subsequent machine learning analysis. Active learning, a technique that selects the most informative samples for the model, could reduce the cost of such a lengthy procedure by achieving better model performance with fewer labelled samples. However, generic batch-mode active learning algorithms are designed for sampling from a single sample pool and thus lack the mechanism to accelerate concurrent experiment execution with multiple data groups in such scientific scenarios. This paper proposes an active learning approach for scientific data with inherent group information, integrating multiple-output quantile regression for uncertainty estimation and combining the diversity of data distribution as a hybrid query method. The proposed method improves the efficiency of concurrent experiments, and the experimental results demonstrate its effectiveness on a suite of material science tasks.

## 1 Introduction

Active learning (AL) approaches are powerful means of optimising predictive model performance relative to the cost of data acquisition (Settles, 2009). These considerations are particularly relevant in scientific domains that may require extensive synthesis, characterisation, experimentation, and/or simulation. These data generation costs are significantly higher than queries to a human oracle, a typical example of an expensive annotation process. Thus, AL is strongly motivated to drive discovery and verification of scientific properties or principles by alleviating budget limitation problems, removing redundancy of experimentation, and eliminating sources of bias (Graff et al., 2021; Smith et al., 2018).

However, even in domains with a significant basis in data-driven research, such as biology (Thornton et al., 2021), biomedical science (Acosta et al., 2022), materials science (Xu et al., 2023), and chemistry (Abolhasani & Kumacheva, 2023), AL implementations may result in inconsistent and inefficient annotation processes. Inefficient annotation, wherein equivalent or superior predictive model performance is achieved with a random sampling strategy, has been noted in certain scientific cases (Stolte et al., 2025; Figueroa et al., 2012). Inconsistent performance of AL (stemming from sensitivity to random initialisation or poor dataset-level robustness) can be severe on small or higher-dimensional datasets (Dong et al., 2024); this instability is a strong barrier to the uptake of AL methods into one-shot scientific workflows.

Underlying constraints in real-world synthesis or experimentation impose structure on data acquisition, for example, grouped synthesis at a particular temperature, catalyst (Xu et al., 2024), pressure (Li et al., 2022), restricted availability of chemical constituents (Mülhopt et al., 2018), high-throughput synthesis (HTS) (Roberts & Owen, 2011). Such constraints are often defined by experimental design and known in advance (Kuznetsova et al., 2024). The input features for the ML model, such as composition, structure, or synthesis conditions, can typically be obtained prior to synthesis from reference databases, computational methods or designed experimental parameters (Rajan et al., 2023; Himanen et al., 2020; Ramprasad et al., 2017;

Hossain et al., 2016). In contrast, the annotation requires synthesising the physical samples under these constraints first, followed by experimental characterisation to get labels (Xu et al., 2023). Typical batch AL has the potential to accommodate such constrained settings by grouping query tasks. However, no current AL framework will accommodate grouped synthesis constraints. In the conventional AL framework, all unlabelled samples are in the same pool, and data selection is globally optimised, with no mechanism to consider underlying data groupings.

Synthesis constraints are ubiquitous. Samples from the same group are generated on the same synthesizer (*e.g.* physical microplate) under the same environmental conditions (*e.g.* temperature, pressure) or constituents (whether chemical or macroscale). Gradual physicochemical variations within the synthesised group create diverse data (Magalhães et al., 2010). This type of data generation with constraints encompasses traditional synthesis and HTS methods. HTS methods process sample groups rapidly, accelerating scientific discovery (Tan et al., 2023; Macarron et al., 2011; Xu et al., 2023). While recent works have applied AL to guide high-throughput experimental and measurement workflows (Noh et al., 2024; Guan et al., 2023; Liu et al., 2022; Kusne et al., 2020; Terayama et al., 2019), the adaptation of AL strategies to the potential practical group-level constraints in such settings is still underexplored. Integrating AL into HTS workflows involves a trade-off. While AL aims to improve efficiency by selecting the most informative samples, such group constraints in HTS may limit the annotation efficiency, potentially reducing the practical benefits of AL in HTS settings. For effective use of AL in these domains, sample selection must also be performed within each synthesis group with a query batch size that matches the synthesizer capacity (defined as constrained AL framework). Otherwise, in the conventional AL framework, selected samples will be dispersed across different synthesis groups, resulting in inefficient use of synthesizer space and longer running time. The capability of multi-sample efficient parallelism in the bathed synthesis process, when combined with AL, will be limited. The examples and comparison of the conventional AL framework and the constrained framework are shown in Figure 1. Assume we select the same number of samples under these two frameworks in one AL iteration, so that the total length of the blue blocks in the two sub-figures is the same. The length of the red dashed square represents the capability of each synthesizers. Since in the conventional AL framework, the selected sample count in some groups likely exceeds the capability of a single synthesizer (G1 in sub-figure (a)), and selected samples count in some other groups are much less than the capability of a synthesizer (G2-4 in sub-figure (a)), which leads running more synthesis process (number of red dashed square) and also

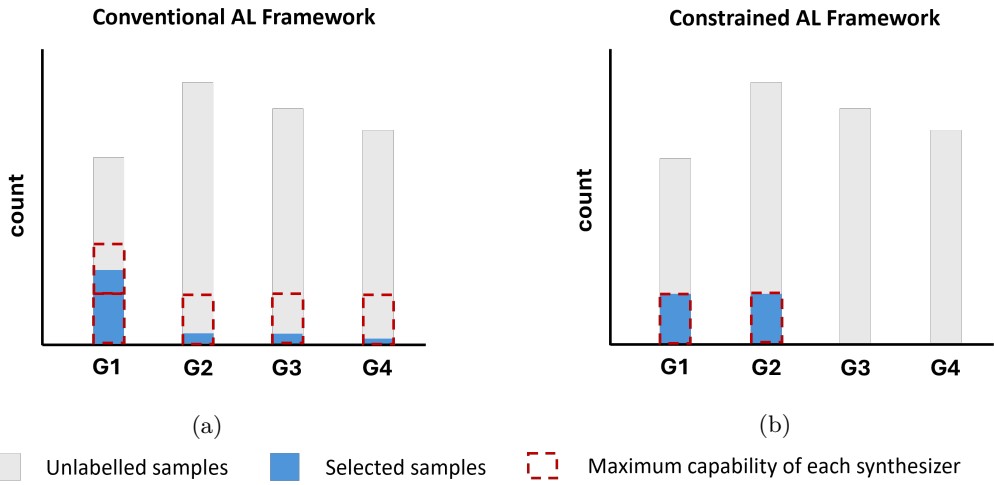

(a)                  (b)

Figure 1: Comparison of sample selection in one AL iteration between the conventional AL framework and the constrained AL framework. The x-axis refers to different data groups, and the y-axis represents the sample counts. If the available synthesisers in this case are two devices, then the conventional AL framework will need to conduct multiple rounds of synthesis processes, but our constrained framework can synthesise all samples using two synthesisers in parallel. The conventional AL framework has synthesis processes where the sample size is below the device's capacity, and it is inefficient and wasteful in real applications.

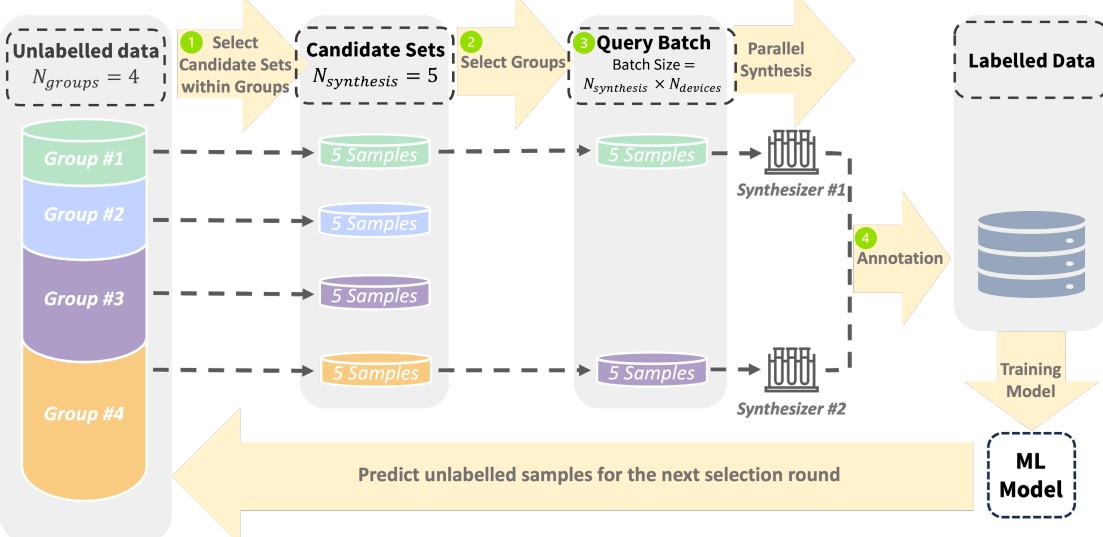

Figure 2: An example shows our constrained AL framework. The unlabelled data comprises $N_{\text{groups}} = 4$ distinct data groups. We assume $N_{\text{devices}} = 2$ synthesizers are available for parallel sample synthesis. In each iteration, the acquisition function selects $N_{\text{synthesis}} = 5$ samples from each group as their candidate set. Then, $N_{\text{devices}}$ groups are selected, and their candidate sets are queried to form a batch of size $N_{\text{synthesis}} \times N_{\text{devices}}$ for synthesis and annotation. Note that Step 3 represents both the selection of the batch and the selected samples, as shown in the grey box.

waste of the synthesizer space each time. But in our constrained framework, we limit the sample selection to a fixed number of groups based on the count of the synthesisers, so that it is much more efficient. In the next AL iteration, when the model is more confident of the samples in G1 and G2, the AL strategy will select samples in G3 and G4. We also have samples in various groups after several AL iterations.

Recent traditional AL algorithms are typically formulated for classification problems (Gal et al., 2017; Ash et al., 2019; Wang et al., 2022), and most of these methods cannot be easily extended to scientific regression tasks (Yarahmadi et al., 2023; Ravi & Desikan, 2023). This distinction is due to the classifier's predictive probability (*e.g.*, the output of the softmax function), which directly provides the model confidence for AL acquisition functions, while regression does not. Furthermore, query strategies for AL are commonly designed based on uncertainty or diversity metrics (Fu et al., 2013; Wang et al., 2022). The uncertainty-based AL methods for regression rely on ensembles (Krogh & Vedelsby, 1994), or are model-specific (Riis et al., 2022), which are not general enough for different application scenarios. Some uncertainty estimation methods designed for deep learning frameworks (Liu et al., 2020; Amini et al., 2020) are not well-suited for scientific tabular data in AL, as they are coupled with deep neural architectures or involve sensitive hyperparameters. AL methods use hybrid selective metrics, which can lead to superior model performance (Shui et al., 2020; Ash et al., 2019), but a hybrid AL method for regression is underdeveloped compared to classification as noted in recent work (Holzmüller et al., 2023).

To address the above limitations and maximise the value of AL in data generation with constraints to accelerate scientific discovery, we propose a new AL regression approach based on hybrid metrics and introduce a constrained AL framework designed for group processes. Specifically, we apply a multiple-output quantile regression model (MQR) with CatBoost (Koenker & Bassett Jr, 1978; Dorogush et al., 2018) to estimate the probability of the prediction intervals for each unlabelled sample. Compared to other methods, MQR offers a flexible alternative for uncertainty estimation in regression that requires no distributional assumptions, exhibits robustness to outliers (Huang et al., 2017) and can be implemented using a wide range of regression models, including linear model (Koenker & Bassett Jr, 1978), neural network (Feldman et al., 2023), tree-based model (Meinshausen & Ridgeway, 2006), and Gaussian Process (Yang et al., 2018), although in this study we focus on tree-based MQR for our experiments. Next, we evaluate the uncertainty for each unla-

belled sample based on the entropy (Shannon, 1948). In each data group, we apply the K-Means++ seeding algorithm (Arthur & Vassilvitskii, 2007) to enhance the diversity of our sample selection. Figure 2 shows an example of our constrained AL framework. The unlabelled data consist of $N_{\text{groups}}$ distinct data groups. We limit the AL selection from $N_{\text{devices}}$ different groups in each AL iteration, where $N_{\text{devices}}$ symbolises the number of synthesizers that can run experiments in parallel, and each of them has a capability of $N_{\text{synthesis}}$ samples. Using the AL acquisition function, we evaluate and identify the ranked top $N_{\text{synthesis}}$ samples from each group as their candidate sets. We then calculate the averaged acquisition scores of samples in each group's candidate set for the group ranking. Finally, from the top-ranked $N_{\text{devices}}$ groups, we select all samples of their candidate sets to annotate in each AL iteration.

Our contributions are: (1). We introduce a framework and a simple AL strategy to accelerate data generation with constraints by limiting sample selection to predefined groups, which guarantees experimentation efficiency under experimental equipment or resource limits. The other AL methods can also be applied to our constrained framework. (2). We leverage MQR to get the probability of the prediction intervals for regression tasks. This is analogous to the role of the classifier's predictive probability in AL. Therefore, it bridges the gap between AL regression and classification methods and allows us to extend some classification-designed AL methods for regression. (3). Experiments demonstrate its effectiveness for scientific tasks. We also provide an analysis of scenarios where our method is not applicable. In addition, it can also be adapted to other applications where the data processing or annotation is group-wise.

## 2 Related Work

The majority of AL approaches for regression can be classified into uncertainty-based and diversity-based query strategies or hybrid methods that combine different aspects of AL metrics.

**Uncertainty and Diversity in AL**. The classical uncertainty-based AL regression method is the Query-by-Committee (QBC) (Krogh & Vedelsby, 1994; RayChaudhuri & Hamey, 1995), in which the uncertainty is calculated as the disagreement among the learners in the ensemble model. The effectiveness of QBC depends on the diversity of learners. Information-theoretic and related variance-based Bayesian AL approaches are often restricted to models that can produce closed-form or accurately approximate predictive or posterior uncertainty, and cannot be readily applied to tree-based models, which are considered in this work (Cohn et al., 1996; Riis et al., 2022). In addition, there are some other uncertainty estimation methods. For example, Gaussian Process (Rasmussen & Williams, 2006) provides posterior variance for uncertainty approximation, but the accuracy depends on selecting the appropriate kernel function and suffers from a cubic training complexity, which might be challenging in practical AL applications. SNGP (Liu et al., 2020) is proposed for large-scale deep neural network architectures and may not be very suited for some smaller tabular data in scientific tasks. Another method, EDL (Amini et al., 2020), relies on some distribution assumptions, and it has limitations in that the regularisation coefficient is sensitive and needs to be fine-tuned. This might hinder its robustness and feasibility in AL.

The diversity-based AL regression methods aim to maximise the minimum distance between unlabelled samples and the labelled set. For example, based on greedy passive sampling (Yu & Kim, 2010), Wu et al. (2019) proposed three schemes of greedy sampling (GS) to characterise the diversity in the feature space (GSx), the output space (GSy), and from both (iGS). Their methods are simple and efficient. However, GSx and iGS sampling are sensitive to the topology information of the feature space, whereas the AL selection of GSy relies on the model accuracy.

**Other Criteria in AL**. Another AL regression method, expected model change maximisation (EMCM), is to select a batch of samples that potentially makes the most significant change (gradient) to the model (Cai et al., 2016). They use Bootstrap to construct an ensemble of models, estimate the predictive distribution of labels, and calculate the gradient of each sample while considering the correlations among the samples. The graph-based AL for regression (Zhang et al., 2020) uses the feature vectors of samples to formulate the query strategy as a bipartite graph optimisation process. The graph nodes are represented as labelled and unlabelled samples to estimate the uncertainty reduction of removing samples from the unlabelled set to the labelled set. The regression tree-based AL (RT-AL) method (Jose et al., 2023a; 2024) selects samples by building a standard regression tree on the labelled set and sampling from leaves, which uses information

from the feature space and the output space. Their extension work QRT-AL (Jose et al., 2023b) is to apply quantile regression, assign weight to different quantile intervals and aim to sample the area in the quantile of interest. This is the first work to use quantile regression on AL selection, but their method and task are completely different from ours. When sampling from the leaves, their method assigns some predefined weights to different quantile intervals to represent the different interest levels on different ranges of target values. It tends to select more samples from their interested ranges. However, our method is to sample across the whole label space used for general scientific discovery purposes. We evaluate the interval probability from the MQR predictions, which differs from the QRT-AL method.

**Hybrid AL Strategies**. Hybrid AL strategies that combine multiple metrics in sampling have become more prevalent in recent years (Ren et al., 2021). For instance, the multiple criteria AL (MCAL) (Demir & Bruzzone, 2014) method considers relevancy, diversity, and density criteria in the sample selection. It uses a two-step procedure based on the SVR model and a clustering approach to select the most representative and informative samples. The inverse-distance based exploration for AL (IDEAL) method (Bemporad, 2023) uses inverse distance weighting (IDW) functions to calculate the uncertainty of samples for informativeness, considers the density function to guarantee representativeness, and uses IDW exploration of unselected areas for diversity. However, the IDEAL method requires setting an appropriate exploration weight parameter. The iterative representativeness diversity maximisation (iRDM) method (Liu et al., 2021) uses the k-means algorithm and the diversity metric of the GSx approach to select samples that consider representativeness and diversity in sample selection. This method does not use any information from the model and is suitable for data with clustering characteristics in the feature space. Besides the above methods, most hybrid AL methods are designed for classification (Lughofer, 2012; Huang & Zhou, 2013; Yang et al., 2015; Ash et al., 2019; Yin et al., 2017; Zhdanov, 2019; Shui et al., 2020; Wu et al., 2021).

**AL in Scientific Applications.** Many studies combine AL and chemistry synthesis to enhance compound-specific screening methods. For example, research has applied Bayesian optimisation (BO) (Snoek et al., 2012; Shahriari et al., 2015) and AL to verify the performance of different acquisition strategies such as greedy, upper confidence bound, and Thompson sampling to accelerate HTS experimentation (Graff et al., 2021). In catalyst synthesis, the expected improvement and predictive variance acquisition functions are used with BO to reduce experimentation time, carbon dioxide footprint, and operating cost (Suvarna et al., 2024). Furthermore, some AL applications exist in materials discovery and design (Kusne et al., 2020; Jablonka et al., 2021; Lookman et al., 2019). However, these studies focus on applying AL methods rather than designing AL methods for specific experimental scenarios, but we design a new AL method suitable for scientific scenarios that have constrained experimentation.

**Quantile Regression.** Quantile Regression (QR) can estimate the conditional distribution in regression problems, using prediction intervals to assess the uncertainty of predictions, and has been widely explored in the past (Koenker & Bassett Jr, 1978; Koenker, 2005) in various domains (Yu et al., 2003; Koenker, 2004). Some recent applied engineering studies use simple QR metrics to calculate the uncertainty in an AL context (Nguyen et al., 2024; 2025), for example, from a confidence interval between two quantiles. MQR can estimate the multiple response variables and formulate a distribution of predictions to yield more probability information and has been studied in (Wen et al., 2017; Gasthaus et al., 2019; Lim et al., 2021). These characteristics of MQR provide an opportunity to formulate a general-purpose uncertainty-based AL strategy.

## 3 Problem and Active Learning Framework

This section presents our AL framework with data generation constraints designed for the scientific group data workflow.

### 3.1 Problem Setting

In pool-based AL, we have a data set $\mathbb{D}$ that is completely characterised in feature space but has incomplete label information. Acquiring new labels requires the additional cost of querying an oracle. This framework applies to a pool-based AL setting in which the unlabelled data exhibit an underlying grouping of samples

in one or more dimensions of the known feature space (*e.g.*, synthesis temperature or chemical constituents). This grouping is intended to relate to a common synthesis setting that necessitates batching processes in laboratories and other scientific scenarios. To emulate this problem, we define an integer input parameter $N_{\text{devices}}$, the number of synthesizers available to generate physical samples simultaneously. There are $N_{\text{groups}}$ data groups that can vary in size and impose a synthesis constraint which restricts us to synthesising batches of size $N_{\text{synthesis}}$ from the same group in each AL iteration.

In pool-based AL, the data set $\mathbb{D}$ is defined for an indexed set of points $\mathbb{P}$ as the union of the unlabelled data set $\mathbb{U}$ and labelled set $\mathbb{L}$, indexed by

$$\mathbb{P} = \mathbb{P}_{\mathbb{U}} \ \cup \ \mathbb{P}_{\mathbb{L}} \tag{1}$$

such that

$$\mathbb{D} = \mathbb{U} \ \cup \ \mathbb{L} = \big\{ \mathbf{x}_i : \ i \in \mathbb{P}_{\mathbb{U}} \big\} \ \cup \ \big\{ (\mathbf{x}_j, y_j) : \ j \in \mathbb{P}_{\mathbb{L}} \big\}, \tag{2}$$

in terms of the $d$-dimensional feature vectors $\mathbf{x} \in \mathbb{R}^d$ and numerical labels $y \in \mathbb{R}$. In our problem setting, categorical or discretised synthesis constraints further arrange the data into $N_{\text{groups}}$ non-overlapping groups indexed by

$$\mathbb{P} = \bigcup_{j=1}^{N_{\text{groups}}} \mathbb{G}_j, \tag{3}$$

where the number of samples in each group can vary. In this framework, at each AL iteration, the labels are queried for a batch containing $N_{\text{devices}} \times N_{\text{synthesis}}$ samples and the corresponding indices are moved from the unlabelled set to the labelled set. The typical batch-mode AL approaches select a batch of samples denoted by the set of indices $\mathbb{B} \subset \mathbb{P}_{\mathbb{U}}$ to label based on an acquisition function $\mathcal{F}$ operating on the unlabelled set of feature data

$$\mathbb{B} = \mathcal{F}\big( \ \mathbb{U}, N_{\text{batch}} \ \big) \tag{4}$$

$$= \mathcal{F}\big( \ \{\mathbf{x}_i : i \in \mathbb{P}_{\mathbb{U}}\} \ , N_{\text{batch}} \ \big), \tag{5}$$

where $N_{\text{batch}}$ is the query batch size. For many AL methods, the acquisition function can be written in terms of a scalar acquisition score $g(\mathbf{x})$, which is a function of the sample's feature vector $\mathbf{x}$. The acquisition function for batch AL then selects the highest valued $N_{\text{batch}}$ scores across all unlabelled feature vectors, so Eq. 4 becomes

$$\mathbb{B} = \underset{|\mathbb{B}|=N_{\text{batch}}}{\arg\max} \ \big\{ g(\mathbf{x}_i) : i \in \mathbb{P}_{\mathbb{U}} \big\}, \tag{6}$$

where $\mathbb{B} \subseteq \mathbb{P}_{\mathbb{U}}$.

## 3.2 Acquisition Scores of Baselines

In this study, we compare our method to the most commonly used AL methods in regression as baselines: Greedy sampling methods (Wu et al., 2019) and Query-by-Committee (QBC) (RayChaudhuri & Hamey, 1995). Greedy sampling is based on maximising the diversity of data sampled by maximising the minimum separating sampled data points. There are three primary variations to greedy sampling: GSx, GSy and iGS, based on which distances are used. GSx maximises the minimum distance of the unlabelled samples to the labelled samples in the feature space. So, for each unlabelled sample $\mathbf{x}_i$, the acquisition score $g(\mathbf{x}_i)$ is the minimum Euclidean distance of $\mathbf{x}_i$ to all the labelled samples $\mathbf{x}_j$ in the feature space,

$$\text{GSx} : g(\mathbf{x}_i) = \min_{j \in \mathbb{P}_{\mathbb{L}}} \|\mathbf{x}_i - \mathbf{x}_j\| . \tag{7}$$

GSy maximises the minimum distance of the unlabelled sample to the labelled samples in the output space, the acquisition score of the unlabelled sample $\mathbf{x}_i$ is

$$\text{GSy} : g(\mathbf{x}_i) = \min_{j \in \mathbb{P}_{\mathbb{L}}} |f(\mathbf{x}_i) - y_j|, \tag{8}$$

in terms of the predicted output value of the base model $f$. We only consider the situation that has one response variable. By combining the GSx and GSy, the iGS method maximises the minimum distance of the

unlabelled sample to the labelled samples from both feature and output spaces, and the acquisition score of the unlabelled sample $\mathbf{x}_i$ is

$$\text{iGS} : g(\mathbf{x}_i) = \min_{j \in \mathbb{P}_\mathbb{L}} \big( \, \|\mathbf{x}_i - \mathbf{x}_j\| \odot |f(\mathbf{x}_i) - y_j| \, \big), \tag{9}$$

which is first to calculate the element-wise product of vectors of the input space and output space distances and then take the minimum value across all labelled samples.

The QBC method uses model disagreement as a measure of model uncertainty to guide sampling. In this approach, multiple sets of labelled data $\mathbb{L}$ are bootstrapped with replacement. A model is trained for each set of labelled data. The committee with $N_{\text{learners}}$ models quantifies disagreement between each model's predicted values $\mathbf{x}_i$, using the variance for each unlabelled sample with acquisition score

$$\text{QBC} : g(\mathbf{x}_i) = \frac{\sum_{j=1}^{N_{\text{learners}}} \Big( f_j(\mathbf{x}_i) - \text{mean}\big(f(\mathbf{x}_i)\big) \Big)^2}{N_{\text{learners}}}, \tag{10}$$

where the $\text{mean}\big(f(\mathbf{x}_i)\big)$ is also defined over the committee.

### 3.3 Constrained AL Framework

Our constrained AL framework is designed for scenarios where unlabelled data are grouped (*e.g.*, synthesis batches). And only a limited number of groups can be selected per AL iteration due to experimental or resource constraints (*e.g.*, number of available synthesizers). Accordingly, the AL selection process must address two levels of decision-making. (1). Which samples are most informative within each group? (2). Which groups to prioritise under a fixed group budget? The difference between the conventional AL framework and our constrained framework is that our selection space is partitioned into groups instead of querying from the entire space of the unlabelled pool. Our selection is constrained to top-ranked groups based on aggregate informativeness. Specifically, in each AL iteration, the framework selects the top $N_{\text{devices}}$ groups and then queries top-scoring $N_{\text{synthesis}}$ samples from each group. Using this constrained framework replicates the scientific synthesis use case and accelerates the batching synthesis (a part of the annotation) process while simultaneously improving efficiency. Any AL query strategies with a defined acquisition score $g(\cdot)$ can be transferred to this framework. Our constrained AL framework shown in Figure 2 is implemented with the following steps:

1. **Select Candidate Sets in Groups**:
   Select $N_{\text{synthesis}}$ samples from each unlabelled data group based on the acquisition function $\mathcal{F}$ to get a candidate batch of indices $\mathbb{B}_j$. $\mathbb{B}_j$ is only defined if the set of unlabelled samples from that group has more than $N_{\text{synthesis}}$ elements. By Eq. 5,

$$\mathbb{B}_j = \mathcal{F}\big( \{\mathbf{x}_i : i \in \mathbb{P}_\mathbb{U} \cap \mathbb{G}_j\}, N_{\text{synthesis}}\big), \tag{11}$$

where $j \in \{1, \cdots, N_{\text{groups}}\}$ and $|\mathbb{G}_j \cap \mathbb{P}_\mathbb{U}| \geq N_{\text{synthesis}}$. If the AL method only defines an aquisition score $g(\mathbf{x}_i)$ for sample selection, Eq. 11 becomes

$$\mathbb{B}_j = \underset{|\mathbb{B}_j|=N_{\text{synthesis}}}{\arg\max} \, \big\{g(\mathbf{x}_i) : i \in \mathbb{P}_\mathbb{U} \cap \mathbb{G}_j\big\}, \tag{12}$$

where $\mathbb{B}_j \subseteq \mathbb{P}_\mathbb{U} \cap \mathbb{G}_j$. The union of the group indices $j$ that have the above candidate sets is denoted as $\mathbb{A}'$.

2. **Select Groups**:
   Based on the number of available synthesizers, determine which groups will be selected by calculating the average value of the acquisition scores $g(\mathbf{x})$ for the unlabelled samples in each group's candidate set. Then rank and select the top $N_{\text{devices}}$ groups,

$$\mathbb{A} = \underset{|\mathbb{A}|=N_{\text{devices}}}{\arg\max} \, \Big\{\text{mean}\big(g(\mathbf{x}_i) : i \in \mathbb{B}_j\big) : j \in \mathbb{A}'\Big\}, \tag{13}$$

where $\mathbb{A} \subseteq \mathbb{A}'$, and $\mathbb{A}$ is the set of group indices selected, and $g(\mathbf{x})$ are the metric values used for ranking samples.

3. **Select Batch Samples**:
   Eq. 4 and 5 are updated to take the candidate set containing $N_{\text{synthesis}}$ samples from each of those groups with the batch indices,

$$\mathbb{B} = \bigcup_{j \in \mathbb{A}} \mathbb{B}_j. \tag{14}$$

   In total, the AL batch size is $N_{\text{synthesis}} \times N_{\text{devices}}$.

4. **Annotate and Update**:
   Samples are synthesised in parallel. Labels $y_i$ are subsequently queried for the batch indices $i \in \mathbb{B}$. The unlabelled and labelled index sets are updated $\mathbb{P}_{\mathbb{L}} = \mathbb{P}_{\mathbb{L}} \cup \mathbb{B}$, $\mathbb{P}_{\mathbb{U}} = \mathbb{P}_{\mathbb{U}} \setminus \mathbb{B}$. Update the base ML model with an updated set of labelled data if required. This step is identical to the traditional batch AL framework.

These steps are repeated for a fixed number of iterations or until a defined stopping criteria are met.

## 4 Active Learning Strategy

Our approach to AL balances selection based on uncertainty estimated in the output space and diversity in the feature space. Our method (**MQR-UD**) leverages hybrid criteria based on multiple-output quantile regression (**MQR**) **U**ncertainty and data **D**iversity to perform AL selection.

### 4.1 Uncertainty Contribution: MQR Acquisition Score

**Multiple-output Quantile Regression**. While ordinary linear regression captures the mean of the response variable, QR explores the conditional distribution by using an asymmetric loss function that penalises overestimation and underestimation differently (Koenker & Bassett Jr, 1978; Steinwart & Christmann, 2011; Romano et al., 2019). This makes QR more robust to outliers and adds value to scenarios when the prediction uncertainty needs to be considered. In our AL method, uncertainty estimation is derived from MQR predictions, which can represent the probability distribution in the output space.

To define this uncertainty contribution to the acquisition score, assume we have a set of $\alpha$ equally spaced quantiles: $\tau_1, \tau_2, \ldots, \tau_\alpha$. We first optimise the MQR using all labelled data $\mathbb{L}$ (as defined in Eq. 2),

$$\hat{\theta} = \arg\min_{\theta} \frac{1}{|\mathbb{P}_{\mathbb{L}}|} \sum_{i \in \mathbb{P}_{\mathbb{L}}} \sum_{k=1}^{\alpha} \left( \tau_k - \mathbf{1}_{\text{condition}} \left( y_i \leq \hat{y}_{ik} \right) \right) \left( y_i - \hat{y}_{ik} \right), \tag{15}$$

where $\hat{\theta}$ is a set of optimised parameters of the MQR, $\hat{y}_{ik} = q_k(\mathbf{x}_i; \theta_k)$ is the predicted value of data sample $\mathbf{x}_i$ by the quantile function $q_k$ at quantile $\tau_k$, and $\mathbf{1}_{\text{condition}}(\cdot)$ refers to the indicator function.

**Predictive Probability in Regression**. Uncertainty estimation for AL in classification can be derived from the predicted probabilities, concretely considered the model confidence (Lewis & Catlett, 1994; Hwa, 2004). However, predictive probability cannot be obtained directly from general regression models. To adapt these uncertainty estimation methods from classification to regression, we employ a simple yet effective density-based probability approach using the multiple outputs of MQR to capture posterior probabilities in prediction intervals for AL regression.

The conditional distribution function of response variable $Y$ given the specific explanatory variable $\mathbf{x}$ is $F(y \mid \mathbf{x}) := P\{Y \leq y \mid X = \mathbf{x}\}$. The quantile function $q_k$ with the quanitle $\tau_k$ is $q_k(\mathbf{x}) := \inf\{F(y \mid \mathbf{x}) \geq \tau_k\}$ (Romano et al., 2019). Let $\{\tau_k\}_{k=1}^{\alpha}$ be a set of $\alpha$ uniformly spaced quantiles, where $\tau_k \in (0, 1)$ for all $k \in \{1, \cdots, \alpha\}$. Corresponding to these quantiles, we have a set of quantile functions $\{q_k\}_{k=1}^{\alpha}$, each trained using the labelled set $\mathbb{L}$. For any sample $\mathbf{x}_i$, it has a set of predictions $\{\hat{y}_{ik}\}_{k=1}^{\alpha}$, the cumulative distribution function (cdf) of the MQR predictions $\{\hat{y}_{ik}\}_{k=1}^{\alpha}$ of $\mathbf{x}_i$ is shown in Figure 3. For any sample $\mathbf{x}_i$, the probability

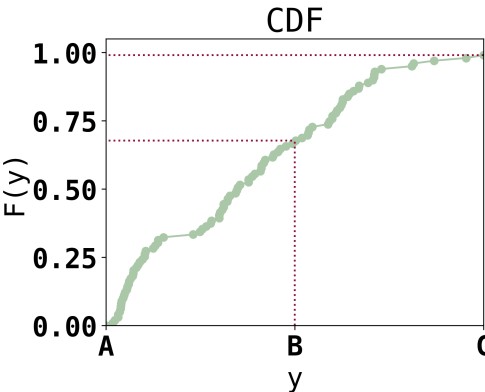

Figure 3: The CDF of the sample predictions of MQR. An example of Pt Nanoparticle data (Barnard et al., 2018)

of $y_i$ falling within an interval $[A, B)$ can be estimated as the proportion of quantile predictions in the interval, where $A, B \in \mathbb{R}$ and $B > A$. The same principle applies to the interval $[B, C]$, where $C$ is the maximum value of predictions, so this boundary is included in the interval.

For a given interval $[A, B)$, the probability of the sample single prediction $y$ being in the interval $[A, B)$ can be defined as $P(A \le y < B) := F(B) - F(A)$, where $F(B) - F(A)$ is the fraction of $F(y)$ within the interval $[A, B)$. Since the quantiles $\tau_1, \tau_2, \cdots, \tau_\alpha$ are uniformly spaced, the predictive probability in the interval $[A, B)$ can be estimated as the proportion of the predicted values failing into the interval

$$P(A \le y < B) \approx \frac{\sum_{k=1}^{\alpha} \mathbf{1}_{\text{condition}}\left(q_k(\mathbf{x}_i; \hat{\theta}_k) \in [A, B)\right)}{\alpha}, \tag{16}$$

where $q_k(\mathbf{x}; \hat{\theta}_k)$ is the prediction value of the $k^{th}$ quantile. The same principle applies to the predictive probability in the interval $[B, C]$, as $P(B \le y \le C)$.

**Prediction Intervals Probability.** Given that the samples in each data group are typically designed according to some common configurations in the same experiment or other synthesis constraints (Shevlin, 2017), they share some common characteristics (Shevlin, 2017). This consistency of the annotation experiment within the same data group allows for meaningful comparison within each group while implying differences among groups that arise from the varying experimental or synthesis conditions. We, therefore, formulate the predictive intervals group-wise to capture details within the data group and ensure interval lengths do not make the probability sparse. In addition, posterior intervals based on individual sample prediction are not comparable because intervals differ among individual samples. The probability of intervals derived from all the unlabelled sample predictions might be sparse due to the big difference in the output space among groups. Thus, we split prediction intervals in each group to characterise the predictive probability for unlabelled samples.

Firstly, based on the feature vectors for each data group $\{\mathbf{x}_i : i \in \mathbb{G}_j\}$, where $|\mathbb{P}_{\mathbb{U}} \cap \mathbb{G}_j| \ge N_{\text{synthesis}}$, let the MQR predictions of all samples across all quantiles $\tau$ be denoted as a matrix $\boldsymbol{A}_j \in \mathbb{R}^{|\mathbb{G}_j| \times \alpha}$. $\boldsymbol{A}_j$ is defined as:

$$\boldsymbol{A}_j = \begin{bmatrix} q_1(x_1; \hat{\theta}_1) & \cdots & q_\alpha(x_1; \hat{\theta}_\alpha) \\ \vdots & \ddots & \vdots \\ q_1(x_{|\mathbb{G}_j|}; \hat{\theta}_1) & \cdots & q_\alpha(x_{|\mathbb{G}_j|}; \hat{\theta}_\alpha) \end{bmatrix}. \tag{17}$$

We then formulate the predictive probability of each sample in the $j^{th}$ group by partitioning $\boldsymbol{A}_j$ into sub-equal intervals based on its prediction interval. We define the total number of intervals as $H$, which is a hyperparameter of the method. The interval width can be calculated as

$$w_j := \frac{\max(\boldsymbol{A}_j) - \min(\boldsymbol{A}_j)}{H}, \tag{18}$$

where $\max(\boldsymbol{A}_j)$ and $\min(\boldsymbol{A}_j)$ are the maximum and minimum values of all the predictions among all samples and quantiles in the $j^{th}$ group. We index intervals from the smallest to largest values with the indices $h = \{1, \cdots, H\}$. The lower and upper boundary of the $h^{th}$ interval in $j^{th}$ group is denoted as

$$
\begin{aligned}
\mathcal{I}^h_{\text{lower}} &:= \min(\boldsymbol{A}_j) + (h-1) * w_j, \\
\mathcal{I}^h_{\text{upper}} &:= \min(\boldsymbol{A}_j) + h * w_j.
\end{aligned}
\tag{19}
$$

For each unlabelled sample $\{\mathbf{x}_i : i \in \mathbb{G}_j \cap \mathbb{P}_\mathbb{U}\}$ in the $j^{th}$ group, we split the predictions $\{\hat{y}_{ik}\}^\alpha_{k=1}$ into $H$ intervals. The total number of predictions of $\mathbf{x}_i$ in the $h^{th}$ interval is defined as

$$
C^h_i := \sum_{k=1}^\alpha \mathbf{1}_{\text{condition}}(\hat{y}_{ik} \in \mathcal{I}^h), \text{ where } \mathcal{I}^h = \begin{cases} [\mathcal{I}^h_{\text{lower}}, \mathcal{I}^h_{\text{upper}}), & h = 1, \cdots, H-1 \\ [\mathcal{I}^h_{\text{lower}}, \mathcal{I}^h_{\text{upper}}], & h = H \end{cases}.
\tag{20}
$$

The predictive probability distribution of sample $\mathbf{x}_i$ across all intervals in the group is denoted as:

$$
\mathcal{P}_i := [P^1_i, \cdots, P^H_i] \in \mathbb{R}^H,
\tag{21}
$$

where $P^h_i = P(y_i \in \mathcal{I}^h \mid \mathbf{x}_i, \mathbb{L}) = \frac{C^h_i}{\alpha}$ represents the probability that $y_i$ falls into the $h^{th}$ interval. We define an acquisition score in terms of the entropy $\mathbb{H}$ (Shannon, 1948) as

$$
g(\mathbf{x}_i) = \mathbb{H}(\mathcal{P}_i) := -\sum_{h=1}^H P^h_i \log P^h_i,
\tag{22}
$$

which has been widely used in AL classification (Wang & Shang, 2014).

## 4.2 Diversity Contribution: K-Means++ Seeded Batching

Typical batch AL acquisition functions (Eq. 4) select the top $N_{\text{batch}}$ ranked samples based on an acquisition function. We, however, use an inherently batched clustering-inspired approach to define the diversity contribution to our acquisition function. The K-Means++ seeding algorithm is an initialisation method for K-Means clustering (Lloyd, 1982) which, in our approach, we use to introduce diversity in data batch selection while saving computational cost compared with the clustering-based methods. Some examples of the K-Means++ seeding algorithm within AL methods are Holzmüller et al. (2023); Ash et al. (2019). We first define the diversity contribution generally (in terms of a general unlabelled data set $\mathbb{U}$ and query batch size $N_{\text{batch}}$), which will be combined with the uncertainty component and the constrained AL framework in the following Section 4.3.

To define a general batch-mode AL acquisition function with the form of $\mathcal{F}(\mathbb{U}, N_{\text{batch}})$ for our diversity contribution, we apply the K-Means++ seeding algorithm (Arthur & Vassilvitskii, 2007; Pedregosa et al., 2011) to select $N_{\text{batch}}$ centre indices $c_n$, where $n = 1, \cdots, N_{\text{batch}}$. The set of centre indices is denoted as

$$
\{c_1, \cdots, c_{N_{\text{batch}}}\} = \text{K-Means++}\big(\mathbb{U}, N_{\text{batch}}\big),
\tag{23}
$$

where $|\mathbb{U}| \geq N_{\text{batch}}$. All unlabelled samples are then assigned to a cluster $\mathbb{C}_n$ based on the minimum Euclidean distance to a cluster centre $c_n$, denoted as

$$
\mathbb{C}_n = \Big\{i : i \in \mathbb{P}_\mathbb{U} \text{ and } \|\mathbf{x}_i - \mathbf{x}_{c_n}\| < \|\mathbf{x}_i - \mathbf{x}_{c_k}\|, \ \forall \, k \in \{1, \cdots, N_{\text{batch}}\} \text{ and } k \neq n\Big\},
\tag{24}
$$

where the index $i$ and indices $\mathbb{P}_\mathbb{U}$ are defined by Eq. 2. By the Eq. 4, we define the batch acquisition function $\mathcal{F}(\mathbb{U}, N_{\text{batch}})$ in terms of a general acquisition score $g(\mathbf{x}_i)$ as selecting one sample from each cluster with the maximum acquisition score,

$$
\mathbb{B} = \mathcal{F}\big(\mathbb{U}, N_{\text{batch}}\big)
\tag{25}
$$

$$
= \Big\{\arg\max\{g(\mathbf{x}_i) : i \in \mathbb{C}_n\} : n = 1, \cdots, N_{\text{batch}}\Big\},
\tag{26}
$$

where $\mathbb{B} \subseteq \mathbb{P}_\mathbb{U}$ and this can be contrasted to the Eq. 6. Within our constrained AL framework, the diversity component is applied separately to each group, defined in Section 4.3.

### 4.3 MQR-UD Hybrid Method

The prior two subsections define our novel acquisition score (Eq. 22) and acquisition function (Eq. 26). We intend to combine these into a hybrid AL method incorporating diversity and uncertainty contributions in an inherently batched approach. The implementation of our MQR-UD AL strategy within the constrained AL framework (Section 3.3) is summarised in Algorithm 1. While steps 3. and 4. of the framework are AL strategy independent, we describe the implementation of Steps 1 and 2 below. **In step 1.** of the framework, both our acquisition score and acquisition function are used to select the candidate batch within available synthesis groups, so Eq. 11 becomes

$$\mathbb{B}_j = \mathcal{F}\big(\{\mathbf{x}_i : i \in \mathbb{P}_{\mathbb{U}} \cap \mathbb{G}_j\}, N_{\text{synthesis}}\big) \tag{27}$$

$$= \Big\{ \arg\max \{\mathbb{H}(\mathcal{P}_i) : i \in \mathbb{C}_k\} : k = 1, \cdots, N_{\text{synthesis}} \Big\}, \tag{28}$$

where $\mathbb{B}_j \subseteq \mathbb{P}_{\mathbb{U}} \cap \mathbb{G}_j$, $j \in \{1, \cdots, N_{\text{groups}}\}$ and $|\mathbb{P}_{\mathbb{U}} \cap \mathbb{G}_j| \geq N_{\text{synthesis}}$. The entropy $(\mathbb{H}(\mathcal{P}_i))$ and clusters $(\mathbb{C}_k)$ defined on the same synthesis group $(\mathbb{P}_{\mathbb{U}} \cap \mathbb{G}_j)$. **The step 2.** of the AL frame, "Select Groups" uses only the acquisition score (Eq. 22) to select $N_{\text{devices}}$ groups by Eq. 13.

The intuition of this design is that within each group, we use hybrid criteria to select informative and diverse samples. The averaged entropy value of candidate samples in each group represents the information density in the group, where higher entropy values indicate higher model uncertainty within the subset. In our constrained framework, selecting groups with higher uncertainty approximates finding data subsets with maximum entropy, constituting group-level entropy maximisation. Since groups are typically assigned based on experimental settings, this framework encourages exploration across top-ranked groups and reduces redundancy.

## 5 Experiments

Our constrained AL framework limits the selection to $N_{\text{devices}}$ groups, where $N_{\text{devices}}$ represents the number of available synthesizers that could generate physical samples. Since in each group we select $N_{\text{synthesis}}$ samples matching the capacity of the synthesizers, our constrained framework can help improve synthesizer utilisation. In scientific discovery tasks, when exploring a larger sample space, the number of groups is usually larger than the number of synthesizers. Thus, applying our constrained AL framework is very valuable since it could save lab resources and improve efficiency in real applications. When comparing baselines within the same constrained AL framework, if the AL method has faster convergence of the model performance, it indicates that it can achieve the same model performance as other AL methods with fewer labelled data, resulting in lower synthesis and annotation costs in the task.

### 5.1 Data

The scientific tabular data sets used in this study are Periodic Graphene Oxide (Barnard et al., 2019), Platinum Nanoparticle (Barnard et al., 2018), Ruthenium Nanoparticle (Barnard & Opletal, 2019a), Palladium Nanoparticle (Barnard & Opletal, 2019b), and Superconductivity (Hamidieh, 2018). The first four datasets are derived from computational simulations and contain multiple labels. We use the most general labels to evaluate the AL performance, which are various energy measurements of the chemical/materials systems in this study. The prediction property of Periodic Graphene Oxide is Fermi Energy. The group information is based on the combination of the amount of C, H, and O components in the materials. The predicted labels of Platinum Nanoparticle, Ruthenium Nanoparticle and Palladium Nanoparticle are Formation Energy. The data are all grouped by synthesis temperature. The lab data Superconductivity is sourced from the UCI Irvine (Dua & Graff, 2017). The predictive label is the superconductor critical temperature. The group information is based on samples with the same chemical constituents (elements) in chemical formulas. Materials with the same chemical constituents require the same raw materials to be produced, and should be reproducible under similar conditions. In this study, we assume that given the same raw materials, a laboratory could reproduce the materials with alternative equipment, or an alternative lab could reproduce the materials with the same equipment. However, we observed that data groups in Superconductivity data

---

**Algorithm 1:** MQR-UD AL Strategy

---

**INPUT:** $\mathbb{U}$: Unlabelled pool; $\mathbb{L}$: Labelled set; $\alpha$: No. of quantiles; $H$: No. of intervals; $N_{\text{devices}}$: No. of groups to be selected at each AL iteration; $N_{\text{synthesis}}$: No. of samples to be selected in the selected group at each AL iteration; $T$: No. of AL iterations; $q$: quantile function;

**OUTPUT:** Updated $\mathbb{L}$

**while** $1 \leq t \leq T$ **do**

  $f = \{q_k(\hat{\theta}_k)\}_{k=1}^{\alpha}$, get the trained MQR model using labelled data $\mathbb{L}$ by Eq. 15;

  $s = 0$, the initial count of available groups in $\mathbb{U}$ that contain more than $N_{\text{synthesis}}$ samples;

  //Step 1: Select Candidate Sets in Groups.

  **for** $j = 1$ **to** $N_{groups}$ **do**

    **if** $|\mathbb{P}_{\mathbb{U}} \cap \mathbb{G}_j| \geq N_{synthesis}$ **then**

      $\boldsymbol{A}_j \leftarrow$ Eq. 17, obtain the MQR prediction matrix for each group;

      $\mathcal{I}_{\text{lower}}^h, \mathcal{I}_{\text{upper}}^h \leftarrow Eq.18, 19$, where $h = \{1, \cdots, H\}$; Construct sub-equal intervals between the maximum and minimum values of $\boldsymbol{A}_j$;

      **for** $i \in \mathbb{P}_{\mathbb{U}} \cap \mathbb{G}_j$ **do**

        $\mathcal{P}_i \leftarrow$ Eq. 20, 21, get the predictive probability of each sample $\mathbf{x_i}$ across all intervals in the group;

      **end**

      $\mathbb{B}_j \leftarrow$ Eq. 22, 27, 28, perform K-means++ within the group, assign each sample to its closest centre, and select the sample index with the largest entropy value in each cluster to form the group's candidate set;

      $s = s + 1$;

    **end**

  **end**

  **if** $s < N_{devices}$ **then**

    | break; Stop the AL cycle since there are not enough available unlabelled data groups.

  **end**

  //Step 2: Select Groups.

  $\mathbb{A} \leftarrow$ Eq. 13, where $g(\mathbf{x}_i) \leftarrow$ Eq. 22. Get the indices of $N_{\text{devices}}$ groups with the highest averaged entropy values within their candidate sets;

  //Step 3: Select Batch Samples.

  $\mathbb{B} \leftarrow$ Eq. 14, query all sample indices from the candidate sets of the selected $N_{\text{devices}}$ groups;

  //Step 4: Annotate and Update.

  Query labels for samples, which indices in $\mathbb{B}$ to form the labelled set $\mathbb{L}^t$ in the $t^{th}$ AL cycle;

  $\mathbb{P}_{\mathbb{L}} = \mathbb{P}_{\mathbb{L}} \cup \mathbb{B}$, $\mathbb{P}_{\mathbb{U}} = \mathbb{P}_{\mathbb{U}} \setminus \mathbb{B}$, $\mathbb{L} = \mathbb{L} \cup \mathbb{L}^t$, $\mathbb{U} = \mathbb{U} \setminus \mathbb{L}^t$;

  $t = t + 1$.

**end**

---

have diverse sizes ranging from 1 to 719 samples. Such inconsistent group sizes make it difficult for the AL setting to set a reasonable $N_{\text{synthesis}}$ to query in each data group, and would reduce efficiency in real applications. Thus, we split this data set into two subsets, Superconductivity-L and Superconductivity-S, respectively. The Superconductivity-L data is filtered by setting the data group sizes to $|\mathbb{G}_j| \geq 100$, and Superconductivity-S is set by the data group size to $|\mathbb{G}_j| < 100$. In this case, we could test a special situation where we have a lot of data groups, but each one is smaller. Superconductivity datasets contain some samples that have the same features $\mathbf{x}$ after characterisation, but they are not identical in the chemical formula. Thus, we keep all these samples to ensure they are consistent with their original data distribution, which this operation is also has been widely tested in previous machine learning tasks (Cui et al., 2021; Ma et al., 2021). The detailed information and the group distributions of each data set can be found in Appendix B.1. We process data sets by removing the mean and scaling to unit variance. The transformation is fitted on the unlabelled pool and applied consistently to the test set. For each data set, we reserve 30% of data testing $\mathbb{D}_{\text{test}}$ and use 70% in an unlabelled pool $\mathbb{U}$.

## 5.2   Model and Training

The design of our method accommodates any regressor that supports quantile loss optimisation. Since the data sets used in this study are tabular, we use the tree-based model for better fitting. Catboost (Prokhorenkova et al., 2018) applies the gradient boosting technique and has been proven to solve the problem of prediction shifts. More importantly, Catboost provides an MQR loss function, enabling a single model to produce multiple quantile predictions by splitting the leaf nodes, significantly accelerating the training process. We use 1000 trees for all experiments and set the early stopping condition as when average training loss reduction over every five 5 epochs cannot be larger than $1 \times 10^{-4}$, training is stopped. We retain the package defaults for other model parameters (Dorogush et al., 2018).

## 5.3   Active Learning

The baseline methods used in this study are introduced in Section 3.2. The original work uses the sequential-mode AL with GSx, GSy, iGS and QBC, while our method uses the batch-mode AL framework, which selects a batch of samples to be labelled and then retrains the model at each AL iteration. Therefore, we adapted GSy, iGS and QBC to the vanilla batch mode in our experiments by selecting a batch of samples based on their ranked scores (distances or prediction variance). The model prediction is the median value of MQR predictions in the baseline methods. GSx does not require any prediction information, so there is no difference between sequential and vanilla batch modes in the selection within each data group. For the baseline methods, we calculate the acquisition scores for each unlabelled sample and then rank the top $N_{\text{synthesis}}$ samples in each data group, computing the averaged acquisition scores of the top $N_{\text{synthesis}}$ samples for group ranking. Then, we select those $N_{\text{synthesis}}$ samples from the top-ranked $N_{\text{devices}}$ groups. For the random baseline, we first choose $N_{\text{devices}}$ different groups randomly, and random selection is applied in each selected data group to select $N_{\text{synthesis}}$ samples. The details of each baseline method can be found in Section 3.2, which is also adapted to our constrained AL framework in Section 3.3.

For Superconductivity-S data, since it contains multiple samples with identical feature vectors $\mathbf{x}$, the K-means++ algorithm would be unable to differentiate between them, leading to ambiguous clustering results. Therefore, in cases where the number of unique samples in each group does not exceed $N_{\text{synthesis}}$, we use only entropy values without doing K-means++ as the function $g(\mathbf{x})$ for these particular groups.

We tested $N_{\text{devices}} = 1$ and 5 in our experiments to represent single and multiple synthesizers scenarios. Since Superconductivity-S has smaller group sizes than other datasets, we use $N_{\text{synthesis}} = 5$ for Superconductivity-S and $N_{\text{synthesis}} = 10$ for other datasets. We use two evaluation metrics, the RMSE and $R^2$ on the separate test set $\mathbb{D}_{\text{test}}$ to evaluate our results. We use the same MQR model at each experiment as the evaluation model among AL methods. The predicted label value $\hat{y}_i$ in RMSE and $R^2$ is the median value of the MQR predictions. All results are from 20 independent trials, and the mean value among 20 experiments for each method is reported. For all experiments, we set the total number of quantiles $\alpha = 99$, and the number of intervals $H = 5$. For the initialisation stage in AL, we first randomly select one data group and randomly query 10 samples in the selected group for all experiments. For the QBC method, since training multiple MQR models is too computationally expensive, we apply $N_{\text{learners}} = 5$ Catboost regression models as the committee models, which is a commonly used hyperparameter in previous research (Bemporad, 2023; Jose et al., 2023b).

## 5.4   Ablation Study

Two sets of ablation studies are included in this work to verify the effectiveness of the method. Firstly, we perform the MQR-UD-reverse experiments, which apply the inverse acquisition scores $g(\mathbf{x})$ to select samples in data groups. It uses the smallest entropy value (Eq. 22) of samples in each cluster to calculate the averaged uncertainty scores to select data groups. The only difference between MQR-UD-reverse and MQR-UD is that we use the smallest entropy values rather than the largest in the MQR-UD. Another ablation study is to evaluate the effectiveness of the hybrid AL metrics. Thus, we remove the K-Means++ seeding algorithm (diversity) and only apply the maximum entropy (uncertainty) as an acquisition function to select data groups. It selects the set of samples with the maximum entropy values in each data group

as the candidate sets and uses the averaged entropy values of candidate sets to select the top groups and then samples. This ablation study verifies the effectiveness of the diversity and uncertainty metrics and names MQR-U. In Figure 8, we compare the MQR-UD, MQR-UD-reverse, MQR-U, and random sampling for analysis. The experimental setting is $N_{\text{devices}} = 5$.

# 6 Results and Discussion

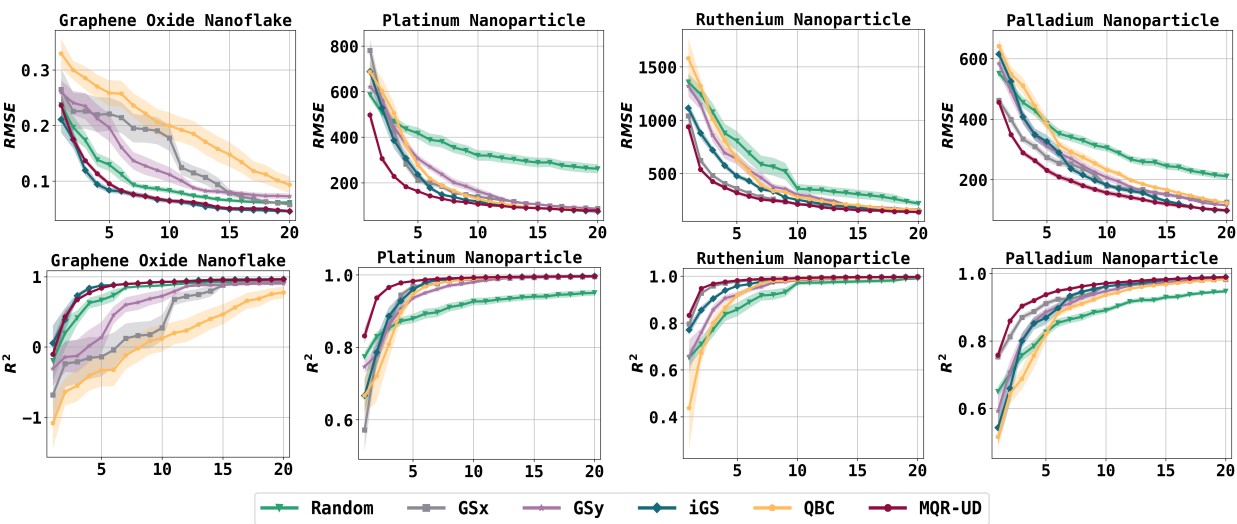

Figure 4: Experimental results of the simulation data. The experimental settings are $N_{\text{devices}} = 1, N_{\text{synthesis}} = 10, T = 20$.

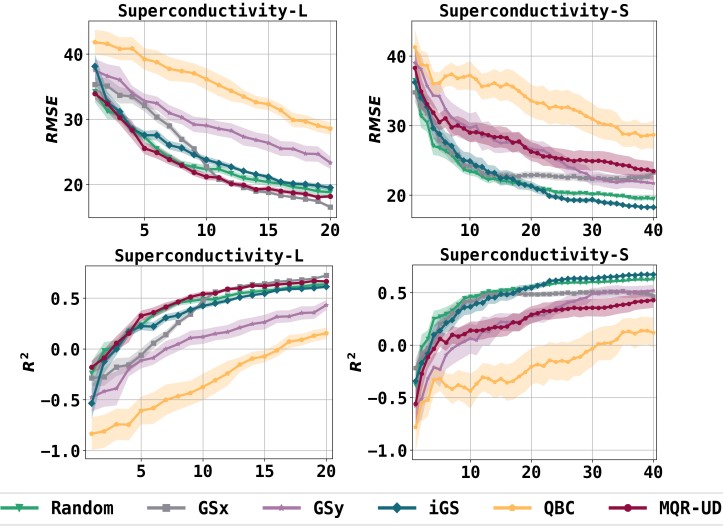

Figure 5: Experimental results of the real data. The experimental settings are $N_{\text{devices}} = 1, N_{\text{synthesis}} = 10, T = 20$ for Superconductivity-L data, and $N_{\text{devices}} = 1, N_{\text{synthesis}} = 5, T = 40$ for Superconductivity-S data.

The main experimental results are shown as Figures 4 to 7. The results of the simulation data are shown in Figure 4 and 6, and the results of real data are shown in Figure 5 and 7. The ablation studies are shown in Figure 8. For all results in these figures, we plot from the first AL iteration, which does not include the AL initialisation step. All AL methods have the same random initialisation in each experiment and the same

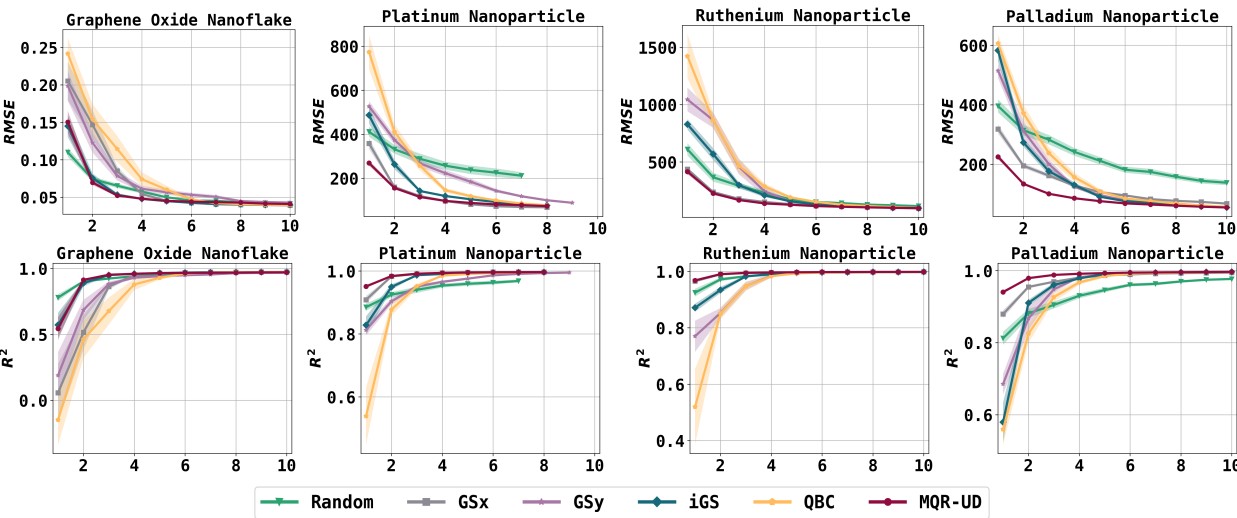

Figure 6: Experimental results of the simulation data. The experimental settings are $N_{\text{devices}} = 5, N_{\text{synthesis}} = 10, T = 10$.

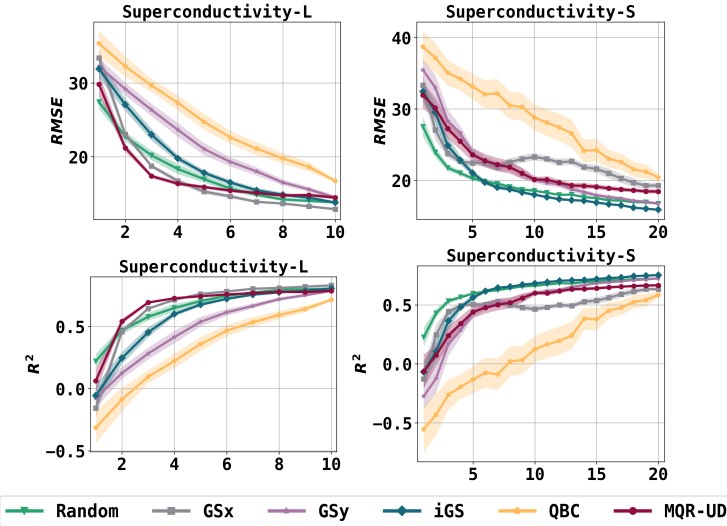

Figure 7: Experimental results of the real data. The experimental settings are $N_{\text{devices}} = 5, N_{\text{synthesis}} = 10, T = 10$ for Superconductivity-L data, and $N_{\text{devices}} = 5, N_{\text{synthesis}} = 5, T = 20$ for Superconductivity-S data.

starting point on the test set in performance. We plot the mean of 20 independent experiments, and the shaded areas represent the standard error.

For simulation data, plots in Figure 4 and 6 demonstrate that our MQR-UD method achieves lower RMSE and higher $R^2$ values with fewer iterations than the other methods overall when $N_{\text{devices}}$ is set to 1 or 5. iGS generally outperforms random sampling of RMSE across most cases, except the Ruthenium Nanoparticle data when $N_{\text{devices}} = 5$. iGS demonstrates comparable performance to our method in some cases, for example, on Graphene Oxide Nanoflake data when $N_{\text{devices}}$ is set to 1 and 5. However, it exhibits less optimal results during early AL iterations in other cases. The performance of GSx varies significantly across data sets, which demonstrates strong results on some data sets while performing markedly weaker outcomes on others. For example, when data has a higher dimension (Graphene Oxide Nanoflake), the diversity measurement from the feature vectors in GSx is not reliable. It leads to worse results in both settings ($N_{\text{devices}}$ is 1 and 5).

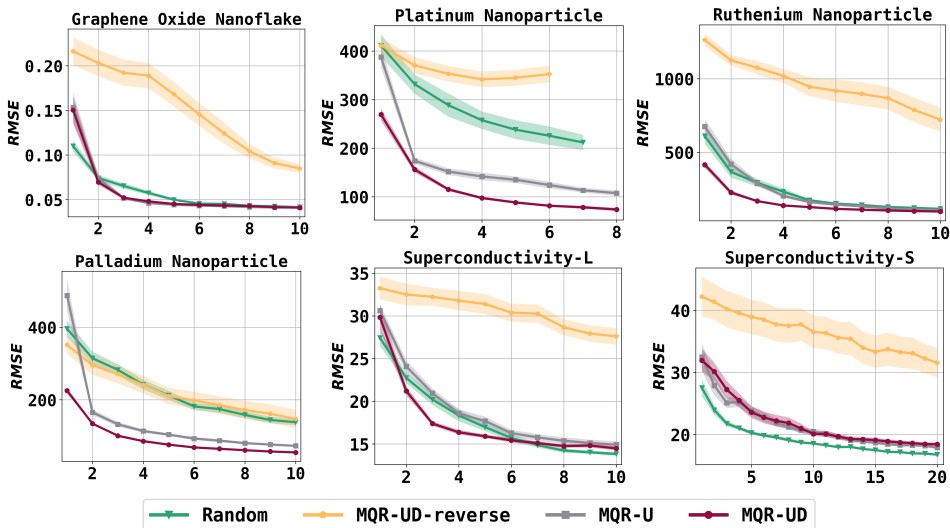

Figure 8: Ablation study results on RMSE metric. The experimental settings are $N_{\text{devices}} = 5, N_{\text{synthesis}} = 5, T = 20$ for Superconductivity-S and $N_{\text{devices}} = 5, N_{\text{synthesis}} = 10, T = 10$ for other data.

However, GSx performed as well as ours on the Ruthenium Nanoparticle data in both settings. In addition, GSy usually performs worse than iGS since it only considers diversity in the output space. QBC always performs worse than the other AL methods and only surpasses GSy in some cases. The possible reason is that the uncertainty estimation is hard because of the lower randomness in the tree-based model compared with the neural networks. Notably, the random sampling sometimes outperforms AL methods, for instance, on the Graphene Oxide Nanoflake when $N_{\text{devices}}$ is 1 and 5, and Ruthenium Nanoparticle with $N_{\text{devices}} = 5$. The $R^2$ results have a similar conclusion to the RMSE results, in which the curves have similar trends. The Platinum Nanoparticle exhibits different stopping points in Figure 6 and 8 due to its relatively smaller group sizes and number of groups. Thus, in the later AL iterations, there are not enough available groups for the query. The AL algorithms terminate when the unlabelled pool contains fewer than $N_{\text{devices}} = 5$ distinct available groups.

For real data results shown in Figures 5, 7, our methods outstrip the other AL methods on RMSE and $R^2$ in both cases $N_{\text{devices}} = 1$ and $N_{\text{devices}} = 5$ of Superconductivity-L data in the early AL iteration, while a little bit worse than GSx in later iterations. Superconductivity-S data has a total of 3863 data groups before the data split, but the number of samples in each group is relatively smaller than that of other data sets. In this case, random sampling and iGS outperform the other AL methods when $N_{\text{devices}} = 1$ and $N_{\text{devices}} = 5$. Our MQR-UD method performs worse and has limitations in this case because of various reasons. Firstly, the small sample sizes within each data group significantly impact our uncertainty estimation approach. This is because our method relies on group boundaries to partition prediction intervals, as defined in Equation 18. Secondly, the cluster-based hybrid method is simple and intuitive, but it has a problem when clusters contain samples with significantly different uncertainty scores; the lower-scoring samples influence the selection of the entire data group. A detailed analysis of these limitations is in the Appendix C.2.

In Figure 8, the ablation study shows that when we use the minimum value of $g(\mathbf{x})$ (entropy value in Eq. 22) to rank groups and select samples, the performance of MQR-UD-reverse is much worse than MQR-UD on RMSE metric. Notably, MQR-UD-reverse even underperforms random sampling across the evaluated data sets, except for Palladium Nanoparticle data, which performs roughly similarly to the random sampling. When we solely use the uncertainty metric (entropy) to query samples, MQR-U consistently underperforms compared to MQR-UD on RMSE, with particularly notable performance gaps observed in the Platinum Nanoparticle, Ruthenium Nanoparticle, and Superconductivity-L data sets. For Graphene Oxide Nanoflake data, the performance of MQR-U is very close to that of MQR-UD since the data selection based on the single uncertainty scores has already introduced diversity in the selection of this dataset. Thus, in this case, the cluster-based selection is not necessary. The performance of Superconductivity-S is the same as the second

point of limitation analysis of this dataset in the above content. The ablation study on the $R^2$ metric is in the Appendix C.1, yielding conclusions similar to those observed for RMSE. The analysis of the hyper-parameter $H$ on the data set Graphene Oxide Nanoflake is shown in Appendix C.3. The scalability of this method is analysed in Appendix C.4; the results show that the selection step of our MQR-UD method only requires several seconds in each AL iteration, which is much cheaper than the MQR model training. But from the scaling results of the model training, the growth rate of training time diminishes with increasing sample size, and memory usage tends to be more stable after some number of samples on most of the datasets. Thus, our method has the potential to scale to large-scale datasets.

Additionally, QR models sometimes require calibration to ensure more accurate coverage rates of prediction intervals, which typically needs a separate labelled calibration set (Romano et al., 2019; Akrami et al., 2022; Feldman et al., 2023). This study uses the median value of MQR predictions for AL evaluation, which is less sensitive than other quantiles. Since AL selection depends on the relative uncertainty values among samples rather than absolute values and accounts for additional annotation costs, calibration is unnecessary for AL selection. However, the calibration step might need to be considered when using our method to predict interval probabilities for other purposes. Future work should also test the flexibility of our method by applying different ML models within the framework. Although we developed our MQR-UD method within the constrained AL framework, this AL strategy can also be applied to general AL frameworks without such constraints. We use the minimum and maximum MQR predictions within each group to construct the prediction interval, as different data groups may share similar output distributions or exhibit relatively distinct and separate ones. But in the future, if we do not use the data that have clearly different groups, we could use the minimum and maximum prediction values of all samples to construct intervals and probability and make this method a general AL strategy. Overall, these findings in this study demonstrate the effectiveness of our hybrid approach, with both the diversity and uncertainty components making meaningful contributions to the overall AL model performance.

Our framework is also meaningful to other domains. In real-world applications, annotation is often conducted in groups across many scenarios. For instance, medical data can be grouped by patient demographics (*e.g.*, age), illness type, or medications, with different clinician groups assigned to their respective specialities. Similarly, in crowdsourcing, tasks are divided into blocks based on annotators' backgrounds and expertise. Our constrained AL framework leverages these natural groupings to improve annotation efficiency by assigning a more balanced number of tasks to different annotators.

## 7 Conclusion

This study bridges AL with the scientific synthesis process to solve data selection within data generation constraint scenarios. We first propose a constrained AL framework to fit this use case and then develop an optimised AL query strategy within that framework. Our framework is broadly applicable. Any AL method with a defined acquisition score can be easily integrated. Our AL regression method applies MQR for uncertainty estimation, and the k-Means++ seeding algorithm is used to increase the data diversity in the selection. Experimental validation demonstrates that our method is competitive with the existing AL methods across various data sets. Therefore, our method is significant for synthesis experimentation in different scientific fields and can help accelerate the lengthy synthesis process of chemical and biological samples, leading to faster scientific discoveries. In future work, the predictive probability derived from MQR can be applied to classification-designed AL methods. Therefore, our work has a profound value for AL in regression and can standardise and narrow the gap between AL methods in classification and regression problems.

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

# A  Notations

- $N_{\text{devices}}$: the number of synthesizers available to generate samples simultaneously.

- $N_{\text{groups}}$: total number of data groups.

- $N_{\text{synthesis}}$: number of samples to be synthesized in each group at each AL iteration.

- $N_{\text{batch}}$: AL query batch size.

- $N_{\text{learners}}$: the number of models in the committee for the QBC method.

- $\mathbb{D}$: data set used for AL.

- $\mathbb{P}$: indices of samples in $\mathbb{D}$.

- $\mathbb{P}_{\mathbb{U}}$: indices of unlabelled samples in $\mathbb{D}$.

- $\mathbb{P}_{\mathbb{L}}$: indices of labelled sample in $\mathbb{D}$.

- $\mathbb{U}$: unlabelled data feature vectors.

- $\mathbb{L}$: labelled data with feature vectors and labels.

- $\mathbb{G}_j$: indices of samples in the data group with group index $j$.

- $\mathcal{F}$: an AL acquisition function.

- $\mathbb{B}$: indices of a batch of samples selected by the AL acquisition function.

- $g(\mathbf{x}_i)$: the AL acquisition score of the sample $\mathbf{x}_i$.

- $f$: the base model.

- $\mathbb{B}_j$: indices of samples in candidate set of the group with group index $j$.

- $\mathbb{A}$: indices of the selected group at each AL iteration.

- $\mathbb{A}'$: the union of the group indices $j$ that have the candidate sets.

- $\alpha$: the total number of quantiles of MQR.

- $\tau_k$: the $k^{th}$ quantile level to be estimated.

- $q_k$: the quantile function at quantile $\tau_k$.

- $\mathbf{1}_{\text{condition}}(\cdot)$: the indicator function.

- $\boldsymbol{A}_j$: the MQR predictions of the $j^{th}$ data group, where $\boldsymbol{A}_j \in \mathbb{R}^{|\mathbb{G}_j| \times \alpha}$.

- $H$: hyperparameter that defines how many intervals (bins) to split in the output space.

- $w_j$: interval width of the $j^{th}$ data group.

- $\mathcal{I}_{\text{lower}}^h$: lower boundary of the $h^{th}$ interval in each data group.

- $\mathcal{I}_{\text{upper}}^h$: upper boundary of the $h^{th}$ interval in each data group.

- $C_i^h$: the total number of predictions from MQR of $\mathbf{x}_i$ in the $h^{th}$ interval.

- $\mathcal{P}_i$: the predictive probability of $\mathbf{x}_i$ from MQR model, $\mathcal{P}_i \in \mathbb{R}^H$.

- $\mathbb{H}(\mathcal{P}_i)$: entropy value of $\mathbf{x}_i$.

- K-Means++$\big(\ \mathbb{D},\ N\ \big)$: K-Means++ seeding algorithm, the input values are dataset $\mathbb{D}$ and the total number of centroids (N) to initialise. The returned values are indices of the N centres.

# B Data Sets Information

## B.1 Data Sets Summary

We preprocess the data sets by removing the columns with missing values, deleting some unnecessary features like ID, and deleting the duplicated samples. In the Superconductivity dataset, duplicated features actually represent different materials in the lab, so we keep all samples the same as the original data. Table 1 shows the data set information after processing. The columns are the number of features, the number of samples, labels are tested in this study, and the information used to split data groups. Figure 9 shows the data group distribution in each data set.

| Dataset | No. of Features | No. of Samples | Label Information | Group by |
|---|---|---|---|---|
| Graphene Oxide Nanoflake | 412 | 1617 | Fermi Energy | C, H, O components |
| Platinum Nanoparticle | 182 | 1299 | Formation Energy | synthesis temperature |
| Ruthenium Nanoparticle | 182 | 2500 | Formation Energy | synthesis temperature |
| Palladium Nanoparticle | 182 | 3996 | Formation Energy | synthesis temperature |
| Superconductivity-L | 82 | 3590 | Critical Temperature | chemical constituents |
| Superconductivity-S | 82 | 17673 | Critical Temperature | chemical constituents |

Table 1: Data sets summary after processing

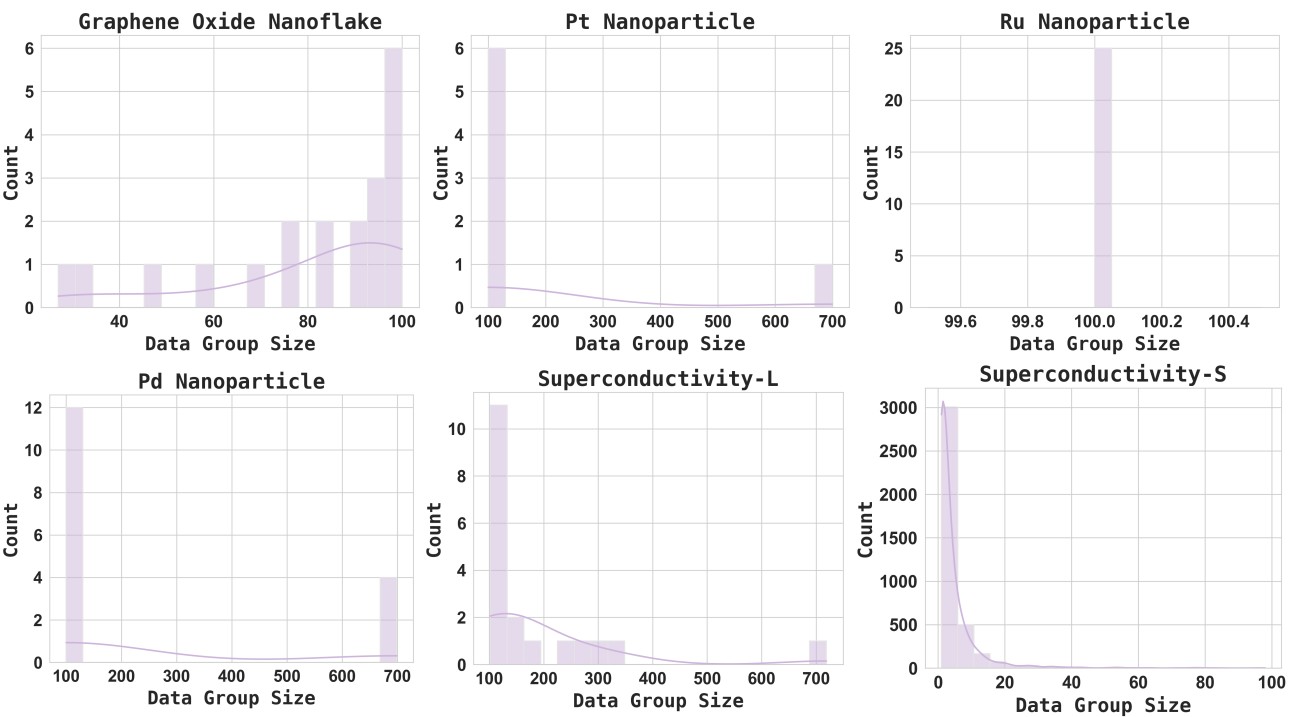

Figure 9: The group distribution.

# C Empirical Study

## C.1 Ablation Study on $R^2$ Metric

In this section, we present the ablation study results of MQR-UD-reverse and MQR-U on the $R^2$ metric (Figure 10). The findings indicate that MQR-UD-reverse significantly underperforms compared to MQR-UD, while MQR-U performs better than MQR-UD-reverse but remains inferior to MQR-UD in most cases. For the

Graphene Oxide Nanoflake dataset, the data selections of MQR-U already encompass the diverse information in feature space, so the performance is quite similar to that of MQR-UD. The comparative performance between MQR-UD and MQR-U aligns with the limitation analysis discussed for the Superconductivity-S dataset in Appendix C.2. Overall, this ablation study shows the effectiveness of both the uncertainty and diversity metrics of the MQR-UD and the design of the acquisition function. It also demonstrates the limitation case of the data selection.

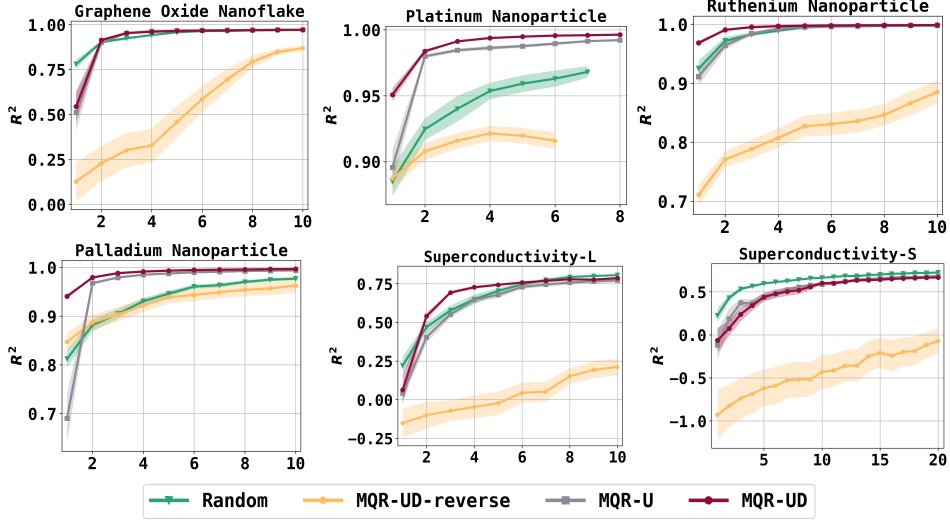

Figure 10: Ablation study results on $R^2$ metric. The experimental settings are $N_{\text{devices}} = 5, N_{\text{synthesis}} = 5, T = 20$ for Superconductivity-S and $N_{\text{devices}} = 5, N_{\text{synthesis}} = 10, T = 10$ for other datasets.

## C.2 Detailed Limitation Analysis on Superconductivity-S data

As shown in Figures 5 and 7, the MQR-UD method underperforms compared to other AL approaches, with the exception of QBC. In contrast, both iGS and random sampling demonstrate excellent performance, surpassing the other methods. Thus, we analyse the reason why our method performs worse on this dataset and discuss the limitations of our method in this section.

Superconductivity-S data has a total of 3863 data groups before data splitting. However, after the splitting, it has averaged 730.4 available data groups (containing samples larger than $N_{\text{synthesis}}$) across 20 trials. This indicates that the size of some data groups is very small. In this case, the uncertainty estimation for each group is not accurate since we use the minimum and maximum of each data group's prediction to split the intervals, which makes the intervals relatively dense. In addition, we use the K-means++ seeding algorithm to introduce mandatory diversity into data selection, but this sometimes causes selection bias that ignores some high-value samples. For example, when a data group has some samples with higher estimated entropy but contains some negative samples with extremely low entropy, the cluster-based method sometimes makes the group ranking consider these negative samples and causes these groups to not be selected because their lower entropy decreases the averaged values within the group. From Figure 11, we can see an example of this case in group 689 (The last line of the Figure). The green shaded area shows that this group contains some samples with very high uncertainty, but the candidate set (red dots) involves the sample with the lower uncertainty values and makes the group have the lower averaged entropy values than other groups and not be selected. Thus, when the K-means++ seeding algorithm cannot provide diverse cluster centres in the feature space, our method might also have unsatisfactory results.

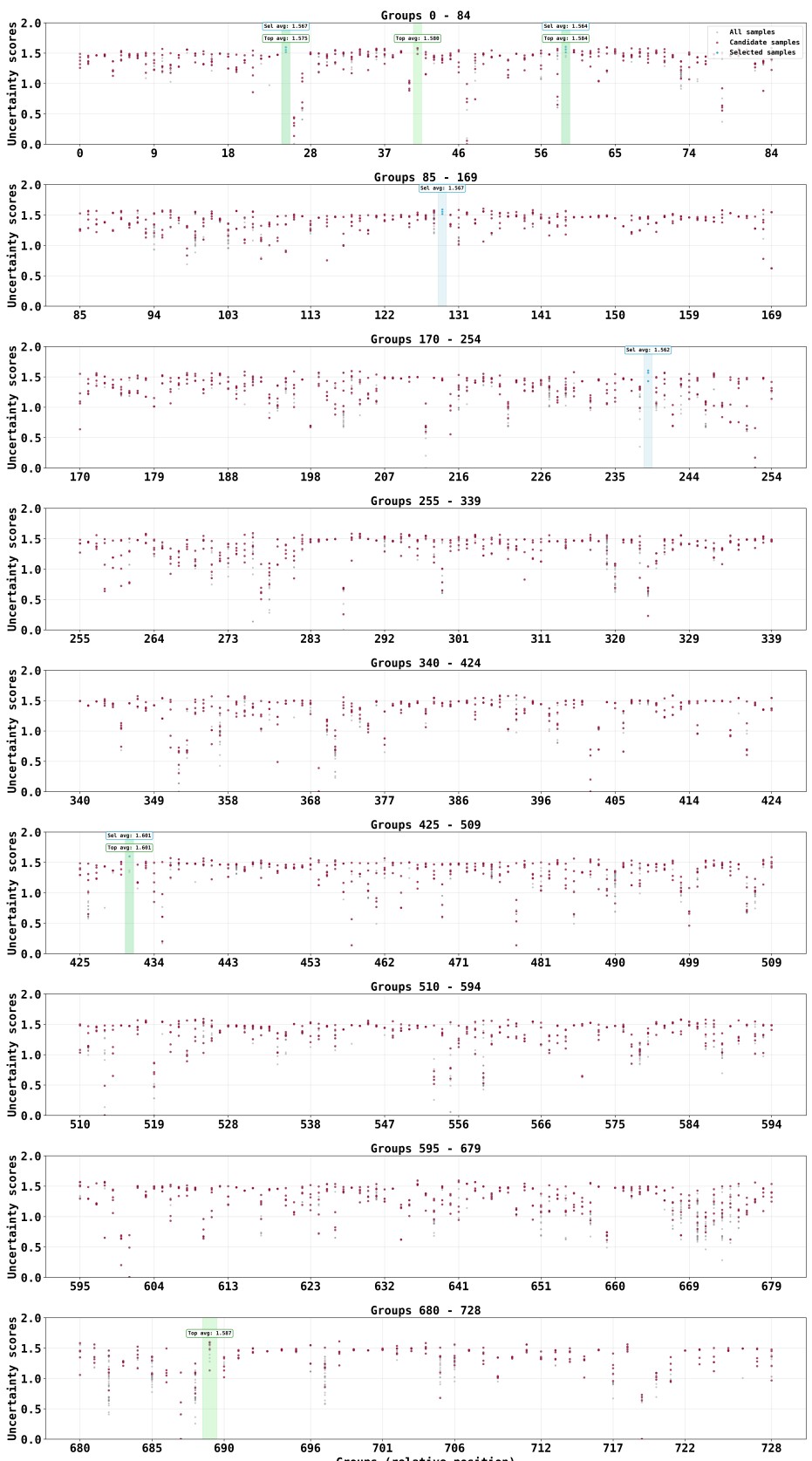

Figure 11: The data selection of the Superconductivity-S data with the setting $N_{\text{devices}} = 5, N_{\text{synthesis}} = 5$. This visualisation represents observations from a single trial in the $5^{th}$ AL iteration. The candidate samples in each group are represented as red dots, while selected samples are displayed as blue dots with blue-shaded areas. The green shaded areas highlight the top-ranked groups containing samples with the highest top $N_{\text{synthesis}}$ entropy values (MQR-U). 'Sel avg' is the averaged entropy value of the selected samples by MQR-UD, and 'Top avg' is the averaged entropy value of the selected samples by MQR-U.

### C.3 Hyper-parameter Analysis

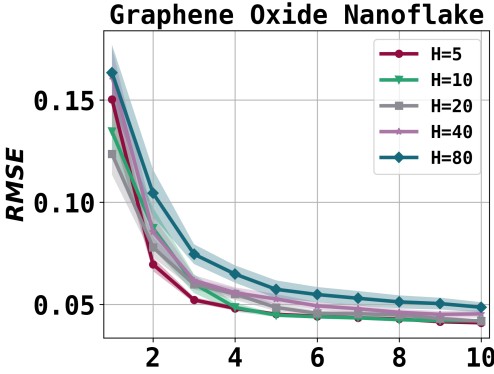

Figure 12: The hyper-parameter analysis of $H$ on Graphene Oxide Nanoflake data, settings are $N_{\text{devices}} = 5, N_{\text{synthesis}} = 10, T = 10$.

In this section, we use the Graphene Oxide Nanoflake data with the settings $N_{\text{devices}} = 5, N_{\text{synthesis}} = 10, T = 10$ to analyse the impact of the hyperparameter $H$ (number of intervals) in this study. From the results shown in Figure 12, we could observe that when increasing $H$ to a larger number, for instance, $H = 40$ and $H = 80$, the uncertainty quantification from interval probability would be affected since the total number of quantiles is $\alpha = 99$ and the interval distribution might be sparse. In these cases, the uncertainty difference of samples might be hard to capture. When setting $H$ as a reasonably smaller value range, the AL performance is relatively better, for example, $H = 10$ in this plot and $H = 5$ in our main results.

### C.4 Computational Cost and Scalability Analysis

All time and memory usage results in this Section were performed on an Apple M3 Max Chip, which has 14 Cores (10 performance and 4 efficiency) with 32GB of Memory. The complexity of our method mainly comes from two parts: the MQR model training and the AL selection step. Thus, we split them separately to evaluate the complexity. To analyse the scaling of the time complexity and the memory usage in model training, we conducted experiments using sample sizes of 100, 500, 1000, 2000, 4000, and 8000 across the datasets. Since the training sets (unlabelled pools) of datasets in our study vary in size, the maximum training size used differs across datasets. The Catboost MQR model we applied is trained with all threads, and all parameters are the same as in our main experiment. The results are the mean and standard deviation values of 20 experiments. For each experiment, we randomly select a subset from the training sets used in our study. The results of running time scaling are shown in Figure 13, and memory usage scaling results are shown in Figure 14.

For the AL selection, we test the running time on two datasets, Graphene Oxide Nanoflake and Superconductivity-S, since they have the highest feature dimensions and largest sample size, respectively. We test the multiple the synthesizers setting that $N_{\text{devices}} = 5$ and the results are shown in Figure 15. In this set of experiments, we set the MQR model with only one tree as the base model, since we only record the AL selection time, excluding the base model training time. The other settings are the same as our main experiments. We report the averaged results of 20 independent experiments with the shaded areas that represent the standard error. Note that in the real application, when training the MQR model with 1000 trees as our main experiments, the AL selection time compared to the results in this Section might be longer since

the computer might have thermal throttling influence, but this depends on the hardware. On the Graphene Oxide Nanoflake dataset, we also perform the scaling experiments of the number of quantiles, using randomly selected 1000 samples to train MQR with $\alpha = [19, 49, 99, 199]$ and run independent 20 experiments. All the other parameters are the same as our main experiment. The result of the mean and standard deviation is shown in Figure 16.

From the scaling results of training time in Figure 13, we could observe that the MQR model exhibits acceptable time scalability on all datasets. The average training time will increase approximately linearly with the growth of the training samples. For most datasets, the growth rate of training time diminishes with increasing sample size. And training with the largest sample size on Superconductivity-S data (8000 samples) can be finished in around 2.23 minutes. In the memory usage plot shown as Figure 14, memory usage increases sharply when the sample size increases from 100 to 500. But it becomes more stable on most of the datasets after 500 samples. Even with the largest sample size on Superconductivity-S data, the peak of memory usage is around 1225MB, which is memory-efficient and scalable to larger datasets. In both Figure 13 and 14, Platinum Nanoparticle data show the linear trend because it is the smallest dataset and has insufficient samples. But it has the same number of dimensions as the Ruthenium and Palladium Nanoparticle datasets. Thus, they should have a similar conclusion with more samples. Figure 16 shows that the training time of the MQR model shows an approximately linear growth trend on Graphene Oxide Nanoflake, which has the highest feature dimensions. This can be used to estimate the running time of the model training in real-world applications with new datasets.

Compared with the model training, we could observe that the AL selection is much quicker. Since in Figure 15, the Random, GSx, GSy and iGS methods all have very short running times. Our MQR-UD method takes longer than theirs, which is less than one second in each AL iteration on Graphene Oxide Nanoflake and less than three seconds in each AL iteration on Superconductivity-S data. The QBC requires training the extra ensemble model that contains five Catboost regressors, which has a longer running time compared with the other AL methods. To compare the AL performance on the same base model, we have an MQR model for QBC to report the evaluation performance on the test set. However, in a real application, QBC does not need another base model and can only use the ensemble model to guide sampling. Thus, it can also save running time for training the MQR base model compared with our experiment in this Section. Compared with the MQR model, the ensemble model training time is much more efficient, since it only requires several seconds, while our MQR requires several minutes. And the selection time of the QBC method is only for calculating the disagreement among learners in the ensemble model, which is very efficient. However, for evaluating all AL methods fairly, we usually report the results on the same model (MQR in this study), which will lead to an extra cost for QBC than in a real application. Thus, we here report in this Section the running time of the real implementation in our research.

In summary, the entropy and K-means++ calculations are much cheaper than the MQR model training in our method. But from the scaling results of the model training, the growth rate of training time diminishes with increasing sample size, and memory usage tends to be more stable after some number of samples on most of the datasets. Thus, our method has the potential to scale to large-scale datasets.

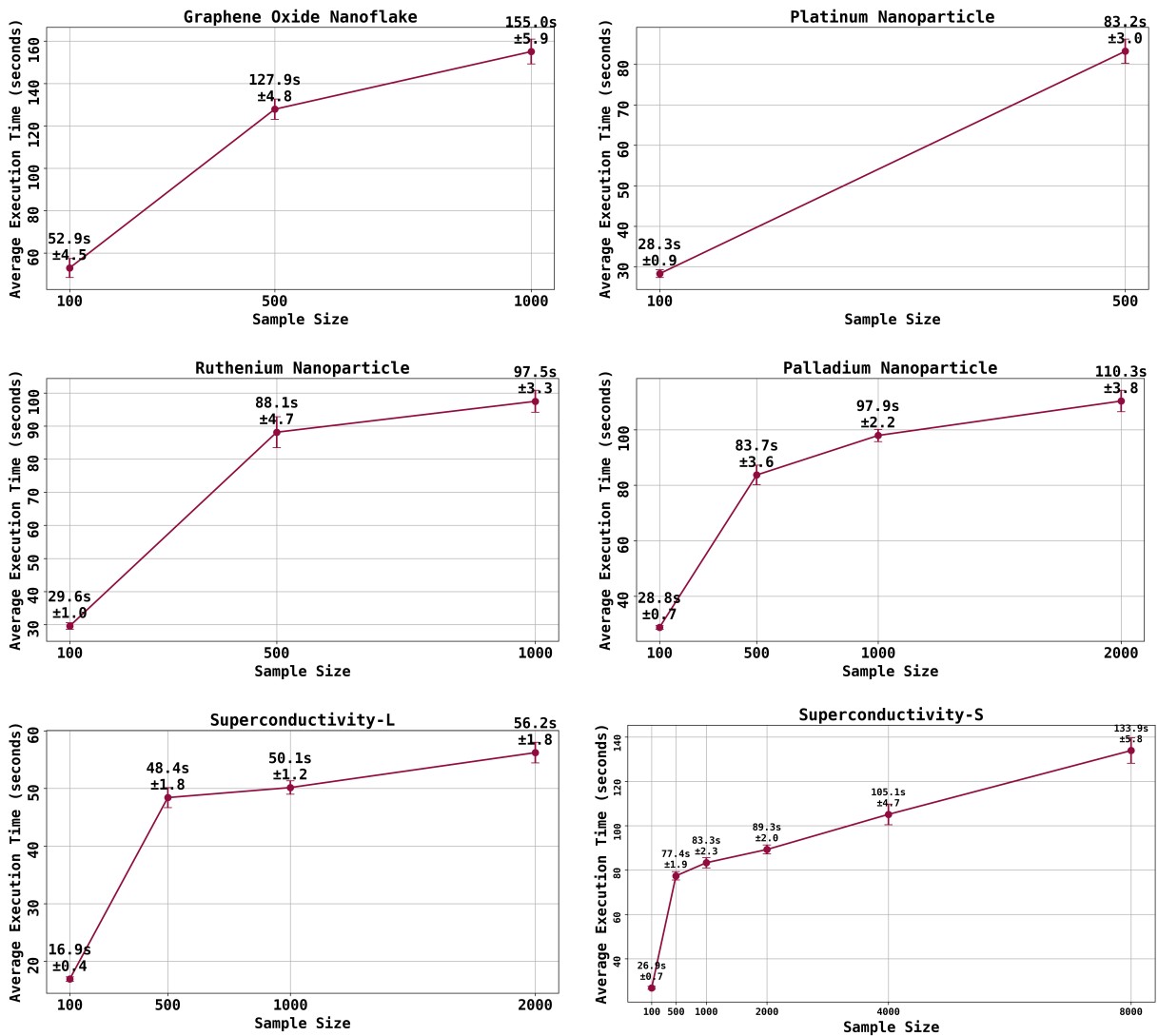

Figure 13: Scaling of Training Time with Training Sample Size

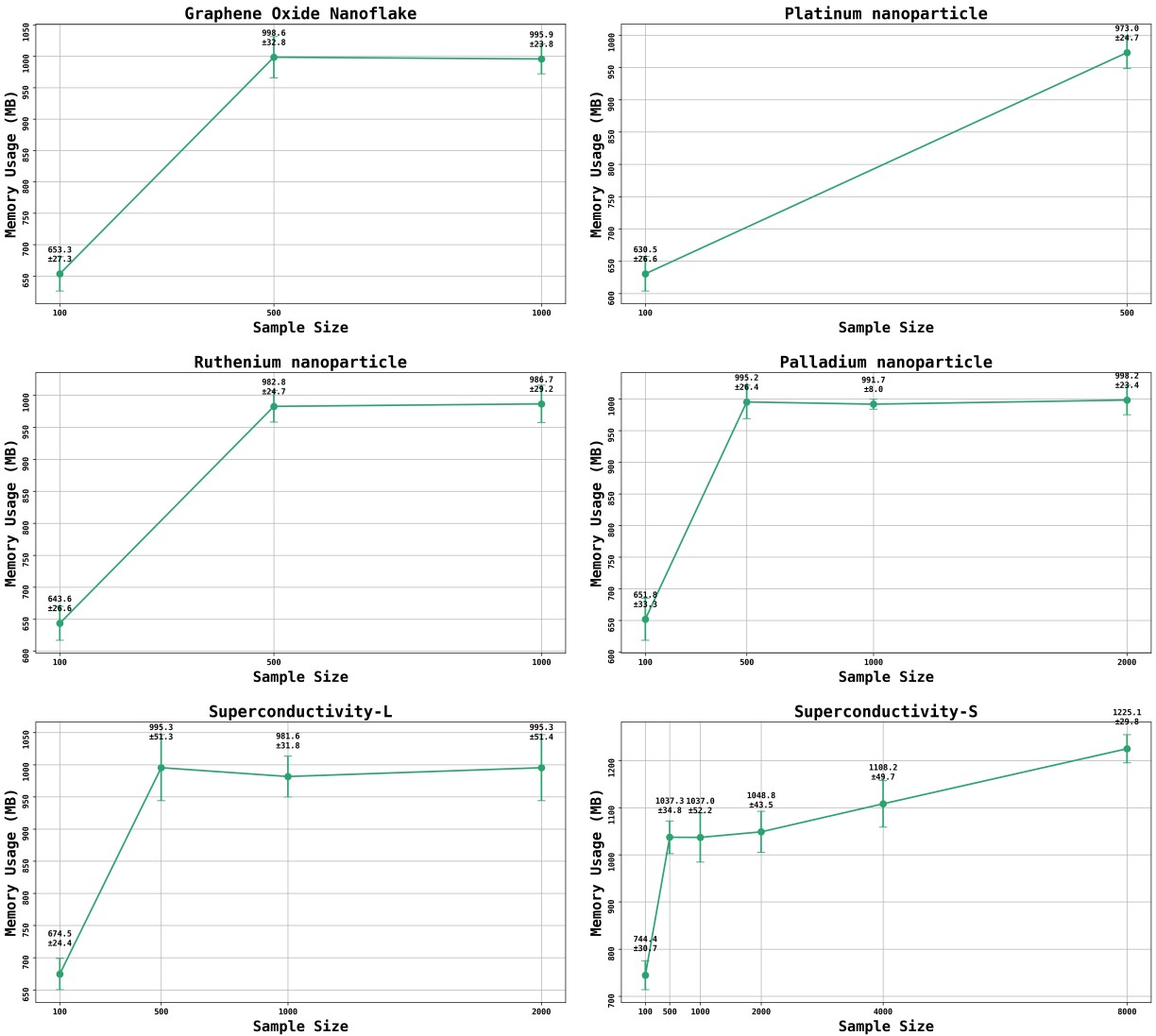

Figure 14: Scaling of Memory Usage with Training Sample Size

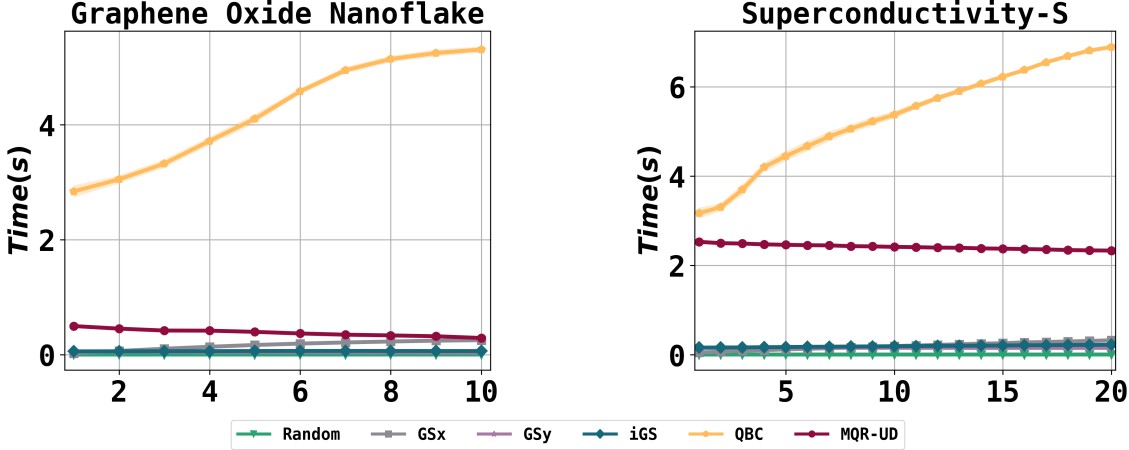

Figure 15: The running time of AL selection. $N_{\text{devices}} = 5, N_{\text{synthesis}} = 10, T = 10$ for Graphene Oxide Nanoflake and $N_{\text{devices}} = 5, N_{\text{synthesis}} = 5, T = 20$ for Superconductivity-S.

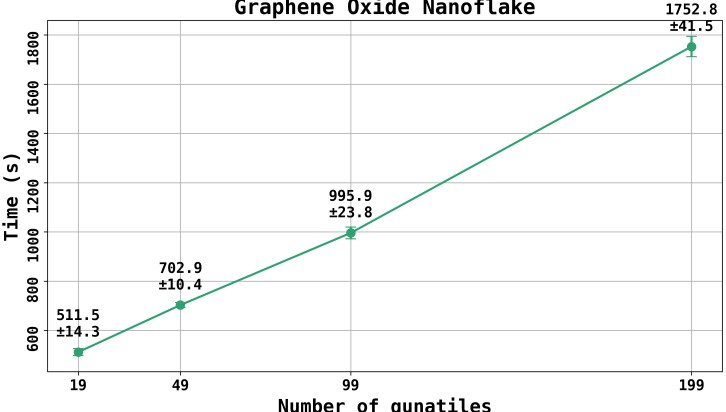

Figure 16: The running time of different numbers of quantiles on Graphene Oxide Nanoflake data.

