# OpenReview forum: "A Hybrid Active Learning Regression Approach for Accelerating Annotation with Data Generation Constraints"
_TMLR — Rejected by TMLR_

### Review · Reviewer_fJaa · 2025-06-27

**Summary Of Contributions:**

This paper introduces an active learning (AL) method for inherently grouped data, such as chemical sample synthesis parameters. The method combines uncertainty (estimated by multi-response quantile regression) and diversity (estimated by the K-Means++ algorithm) in its query strategy. Its efficacy is demonstrated through comparison with existing AL methods and an ablated variant in five tabular computational/experimental materials science datasets.

**Audience:**

Yes

**Broader Impact Concerns:**

None.

**Claims And Evidence:**

No

**Requested Changes:**

### Content issues
- The claim “a systematic study on adapting AL to HTS has yet to be conducted” in the Introduction is not accurate. Many prior works have explored the direction, e.g., [[1]](https://doi.org/10.1038/s42256-022-00460-0); [[2]](https://doi.org/10.1038/s41467-020-19597-w); [[3]](https://doi.org/10.1103/PhysRevMaterials.3.033802). Another point worth discussing is the tradeoff between **rapid** HTS and **selective** AL-guided synthesis.
- The discussion on “uncertainty methods designed for regression” (in Introduction and Sec. 2) doesn’t cover some widely used methods, such as Gaussian Process (GP) and variants, and some model-agnostic plug-ins for deep learning models, including [SNGP](https://www.jmlr.org/papers/v24/22-0479.html) and evidential DL. What are some edges of MQR over them?
- Sec. 5.1 mentions that the whole dataset is normalized. This would probably leak some information from the test set to the training set (unlabeled pool). A justification or revision is needed here.

### Clarity issues
- In the Introduction, where “hybrid metrics” is first mentioned, a brief explanation should follow for better understanding.
- In Sec. 3, the formulas are not clear. The range where acquisition scores are evaluated is critical for understanding AL strategies but not clearly indicated by subscripts of $\arg\max$.
- The meaning of “constraint” in this paper’s context is unclear. Does it mean that some regions of the experimental condition space are infeasible? Would grouping completely address the issue, or does feasibility need to be discovered in the AL process? I suggest adding a definition with examples.

**Strengths And Weaknesses:**

### Strengths
- The paper’s problem formulation shows some practical considerations in scientific scenarios, e.g., grouping of synthesis conditions and efficient parallelism.
- The experimental results are comprehensive, outlining the (in)applicability of the proposed method.

### Weaknesses
- The clarity of writing should be improved (see Requested Changes).
- The discussion of prior works should be more comprehensive and objective (see Requested Changes).

---

> ### Author Response · Authors · 2025-08-11
> **Comment to Reviewer fJaa**
>
> We sincerely appreciate the reviewer’s valuable suggestion. We will address the questions point by point.
>
> **Content issues:**
> 1. We have changed “a systematic study on adapting AL to HTS has yet to be conducted” to: “While recent works have applied AL to guide HTS [1-5], the adaptation of AL strategies to the practical group-level constraints of HTS is still underexplored. ” to clarify this point. Yes, many HTS studies have applied AL strategies to reduce annotation costs. However, these approaches typically assume a global selection setting, where the most informative samples are chosen from the entire search space. Thanks for pointing out the point of trade-off in HTS and AL. We have added several sentences to explain this: “Integrating AL into HTS workflows involves a trade-off. While AL aims to improve efficiency by selecting the most informative samples, such group constraints in HTS may limit the annotation efficiency, potentially reducing the practical benefits of AL in HTS settings.”
>
> 2. We have added these uncertainty estimation methods in the introduction: “Some uncertainty estimation methods designed for deep learning frameworks  [6-7] are not well-suited for scientific tabular data in AL, as they are coupled with deep neural architectures or involve sensitive hyperparameters.” and also discussed them in detail in the Related Work: “In addition, there are some other uncertainty estimation methods. For example, Gaussian Process[8] provides variance for uncertainty approximation, but the accuracy depends on selecting the appropriate kernel function, which might be challenging in practical AL applications. SNGP[6] is proposed for large-scale deep neural network architectures and may not be very suited for some smaller tabular data in scientific tasks. Another method, EDL[7], relies on some distribution assumptions, and it has limitations that the regularisation coefficient is sensitive and needs to be fine-tuned. This might hinder its robustness and feasibility in AL.”
> To explain the applicability and edge of MQR we have added to the following content of the Introduction “Compared to other methods, MQR offers a flexible alternative for uncertainty estimation in regression that requires no distributional assumptions, exhibits robustness to outliers[9] and can be implemented using a wide range of regression models, including linear models[10], neural networks[11], tree-based models[12], and Gaussian Process[13].”
>
> 3. We performed normalisation using only the training set statistics and applied the same transformation to the test set. Therefore, no data leakage occurred in this process. We have added “The transformation is fitted on the unlabelled pool and applied consistently to the test set.” to clarify this.
>
> **Clarity issues:**
> 1. We have added an explanation in the Introduction: “AL methods use hybrid selective metrics, which can lead to superior model performance…”.
>
> 2.  2. We have added the range of the argmax results in Sections 3 and 4, respectively.  For example,
> $$
> \\mathbb{B} = \\arg\\max_{|\\mathbb{B}|=N_{\\mathrm{batch}}}\\left\\lbrace g(x_i): i\\in P_U \\right\\rbrace,\ \text{where } \\mathbb{B}\\subseteq P_U .
> $$
> The other changes are highlighted in our revised manuscript.
>
> 3. We added: “Such constraints are often defined by experimental design and known in advance. The input features for the ML model, such as composition, structure, or synthesis conditions, can typically be obtained prior to synthesis from designed experimental parameters, reference databases, or computational simulations. In contrast, the annotation requires synthesising the physical samples under these constraints first, followed by experimental characterisation to get labels.” in the Introduction, after several examples of constraints in science. The constraints do not necessarily indicate infeasibility in some areas of the condition space and are independent of the AL process. Since our previous expression is misleading: “Typical batch AL can address the above constraints by grouping query tasks.” We have changed this sentence to “Typical batch AL has the potential to accommodate such constrained settings by grouping query tasks” for a better expression.
>
> [contd.]

---

> > ### Author Response · Authors · 2025-08-11
> > **Comment to Reviewer fJaa**
> >
> > [contd.]
> >
> > **References**
> >
> > [1]. Juran Noh, Hieu A Doan, Heather Job, Lily A Robertson, Lu Zhang, Rajeev S Assary, Karl Mueller, Vijayakumar Murugesan, and Yangang Liang. An integrated high-throughput robotic platform and active learning approach for accelerated discovery of optimal electrolyte formulations. Nature Communications, 15(1):2757, 2024.
> >
> > [2]. Xiaoyu Guan, Zhongnian Li, Yueying Zhou, Wei Shao, and Daoqiang Zhang. Active learning for efficient analysis of high-throughput nanopore data. Bioinformatics, 39(1):btac764, 2023.
> >
> > [3]. Yongtao Liu, Kyle P Kelley, Rama K Vasudevan, Hiroshi Funakubo, Maxim A Ziatdinov, and Sergei V Kalinin. Experimental discovery of structure–property relationships in ferroelectric materials via active learning. Nature Machine Intelligence, 4(4):341–350, 2022.
> >
> > [4]. A Gilad Kusne, Heshan Yu, Changming Wu, Huairuo Zhang, Jason Hattrick-Simpers, Brian DeCost, Suchismita Sarker, Corey Oses, Cormac Toher, Stefano Curtarolo, et al. On-the-fly closed-loop materials discovery via bayesian active learning. Nature communications, 11(1):5966, 2020.
> >
> > [5]. Kei Terayama, Ryo Tamura, Yoshitaro Nose, Hidenori Hiramatsu, Hideo Hosono, Yasushi Okuno, and Koji Tsuda. Efficient construction method for phase diagrams using uncertainty sampling. Physical Review Materials, 3(3):033802, 2019.
> >
> > [6]. Jeremiah Liu, Zi Lin, Shreyas Padhy, Dustin Tran, Tania Bedrax Weiss, and Balaji Lakshminarayanan. Simple and principled uncertainty estimation with deterministic deep learning via distance awareness. Advances in neural information processing systems, 33:7498–7512, 2020.
> >
> > [7].Alexander Amini, Wilko Schwarting, Ava Soleimany, and Daniela Rus. Deep evidential regression. Advances in neural information processing systems, 33:14927–14937, 2020.
> >
> > [8]. Carl Edward Rasmussen and Christopher K. I. Williams. Gaussian Processes for Machine Learning. MIT Press, 2006. URL http://www.gaussianprocess.org/gpml/.
> >
> > [9]. Qi Huang, Hanze Zhang, Jiaqing Chen, and MJJBB He. Quantile regression models and their applications: A review. Journal of Biometrics & Biostatistics, 8(3):1–6, 2017.
> >
> > [10]. Roger Koenker and Gilbert Bassett Jr. Regression quantiles. Econometrica: journal of the Econometric Society, pp. 33–50, 1978.
> >
> > [11]. Shai Feldman, Stephen Bates, and Yaniv Romano. Calibrated multiple-output quantile regression with representation learning. Journal of Machine Learning Research, 24(24):1–48, 2023.
> >
> > [12]. Nicolai Meinshausen and Greg Ridgeway. Quantile regression forests. Journal of machine learning research, 7(6), 2006.
> >
> > [13]. Yandong Yang, Shufang Li, Wenqi Li, and Meijun Qu. Power load probability density forecasting using gaussian process quantile regression. Applied Energy, 213:499–509, 2018.

---

### Review · Reviewer_nodK · 2025-07-09

**Summary Of Contributions:**

The paper proposes a novel active learning framework for regression tasks, specifically designed for real-world experimental settings where data samples are generated in groups due to physical or chemical synthesis constraints. The proposed method, referred to as constrained active learning (constrained AL), selects candidate samples within each group based on an acquisition function, ranks the groups according to their informativeness, and then selects a batch of samples from the top-ranked groups for annotation. The framework accommodates different acquisition strategies, including uncertainty-based and diversity-based functions. The method is evaluated on six datasets, and the results demonstrate competitive performance in minimizing both RMSE and R^2, confirming its effectiveness in constrained scientific settings.

**Audience:**

Yes

**Broader Impact Concerns:**

N

**Claims And Evidence:**

Yes

**Requested Changes:**

Discuss Scalability and Efficiency
The authors are encouraged to discuss the computational complexity of the method, particularly with regard to training multiple quantile regressors and computing entropy over group-specific intervals. Providing commentary or empirical profiling on runtime and memory usage would be valuable for practitioners considering deployment on large-scale datasets.



Clarify Generalizability
While the method is well-suited for synthesis-based scientific tasks, it would be helpful to discuss its applicability to other domains where group-wise annotation may arise, such as medical imaging, crowdsourcing, or behavioral studies.

Handling Grouping When Not Given
The current framework assumes that the number and composition of groups are predefined. However, in many domains, datasets may not naturally come with group labels. The authors are encouraged to discuss how the method could be adapted in such scenarios—e.g., by forming groups based on clustering or feature-based heuristics, and then evaluating whether such synthetic groupings yield performance gains compared to standard baselines. Even a qualitative discussion or small-scale experiment on a common benchmark dataset could significantly strengthen the paper.

**Strengths And Weaknesses:**

Limited Model Flexibility
While the acquisition strategy is adaptable, the flexibility of the overall methodology to different base learners is not explored. The experiments rely solely on decision tree-based models (CatBoost with MQR). It would be valuable to see whether the framework maintains its performance when applied to other regression models such as linear regression or Gaussian processes.

Lack of Theoretical Analysis
Although the empirical results are convincing, the paper would benefit from some theoretical justification or insight into why the hybrid acquisition strategy is expected to perform well under constrained settings. This is not a critical flaw, but such analysis could significantly strengthen the contribution.

Scalability Concerns
The use of high number of  quantiles in MQR and group-wise entropy computations may pose scalability challenges for large datasets or frequent active learning iterations. The paper does not discuss computational cost or runtime, which would be useful for practitioners.

Performance Variability and Robustness
In some datasets—particularly Superconductivity-S—the proposed method does not consistently outperform simpler baselines like random sampling or iGS. This raises questions about robustness. It would help to analyze whether the performance drop is due to small group sizes, high data imbalance, or few groups. Furthermore, while results on the simulated datasets are promising, the two real-world datasets do not fully reveal the strengths or limitations of the proposed framework, and further discussion would be helpful.

---

> ### Author Response · Authors · 2025-08-11
> **Response to Reviewer nodK**
>
> We sincerely appreciate the reviewer’s valuable suggestion. We will address the questions point by point.
>
> **Weaknesses**
>
> 1. While our current implementation uses CatBoost with MQR for its strong empirical performance and native quantile loss support, the proposed acquisition strategy works with many other models. We have added some examples in the Introduction. “Compared to other methods, MQR offers a flexible alternative for uncertainty estimation in regression that requires no distributional assumptions, exhibits robustness to outliers[1] and can be implemented using a wide range of regression models, including linear model [2], neural network[3], tree-based model[4], and Gaussian Process[5].” So it can be readily extended to other regressors. Compared to Gaussian Process Regression (GPR),  tree-based models require no input assumptions and natively support multiple quantile loss functions in CatBoost, while GPR requires the use of appropriate likelihood functions to accommodate MQR and also needs to select the proper kernel functions, so it is not practical in AL. Compared to linear regression, tree-based models are generally more robust to outliers and can naturally capture non-linear relationships. In scientific problems, especially for tabular data,  the tree-based models are widely used [6], [7], so we apply the tree-based model in our experiments.
>
> The design of our method accommodates any regressor that supports quantile loss optimisation. We recognise this as a valuable research direction and plan to explore broader model classes (e.g., Gaussian Processes, neural networks) in future work. We have added this to Section 5.2 and the end of Section 6: “The design of our method accommodates any regressor that supports quantile loss optimisation. ” and “Future work should also test the flexibility of our method by applying different ML models within the framework.”
>
> 2. We have added this explanation to Section 4.3. “The intuition of this design is that within each group, we use hybrid criteria to select informative and diverse samples. The averaged entropy value of candidate samples in each group represents the information density in the group, where higher entropy values indicate higher model uncertainty within the selected subspace. In our constrained framework, selecting groups with higher uncertainty approximates finding data subsets with maximum entropy, constituting group-level entropy maximisation. Since groups are typically assigned based on experimental settings, this framework encourages exploration across top-ranked groups and reduces redundancy.” And the ablation study also proves that the two AL metrics in our method are effective.
>
>
> 3. Scalability analysis of time and memory usage has been added to Appendix C.4, and the conclusion is: The scalability results show that the selection step of our MQR-UD method only requires several seconds in each AL iteration, which is much cheaper than the MQR model training. But from the scaling results of the model training, the growth rate of training time diminishes with increasing sample size, and memory usage tends to be more stable after some number of samples on most of the datasets. Thus, our method has the potential to scale to large-scale datasets.
>
> 4. In fact, we have already conducted a detailed limitation analysis of Superconductivity-S in Appendix C.2  of the original submission and also had cross-references to this analysis in the main text. We expect our method to perform poorly on the superconductivity-S, since it is a constructed subset from the whole dataset, designed to test the bounds of this approach. In our MQR-UD, we use the prediction values of each group to calculate the probability. This dataset contains many small groups, and it does not fit the usual prescribed use case. The performance on Superconductivity-L compared to Superconductivity-S supports this.
>
> **Requested Changes**
> 1. We have added the computational complexity about time and memory usage into Appendix C.4. And also conclude in the Discussion.
>
> 2. We have added this discussion to Section 6:
> “Our framework is also meaningful to other domains. In real-world applications, annotation is often conducted in groups across many scenarios. For instance, medical data can be grouped by patient demographics (e.g., age), illness type, or medications, with different clinician groups assigned to their respective specialities. Similarly, in crowdsourcing, tasks are divided into blocks based on annotators' backgrounds and expertise. Our constrained AL framework leverages these natural groupings to improve annotation efficiency by assigning a more balanced number of tasks to different annotators.“
>
> [contd.]

---

> > ### Author Response · Authors · 2025-08-11
> > **Response to Reviewer nodK**
> >
> > [contd.]
> >
> > 3. We thank the reviewer for raising this point. However, we respectfully disagree that since our pipeline specifically relates to synthesis constraints, which are common in science. Other groupings are less relevant to this pipeline unless they have a meaningful impact on batching. For example, as we discussed in the last question, data is grouped by patient demographics or crowdsourcing, but the data groups are highly related to the annotators. This is predefined as well. If data annotation cannot be naturally processed in batches, this framework is not applicable. If we do a clustering-based algorithm to form groups, the downstream annotation is likely not relevant to these groups, and that is not meaningful. Thus, connecting the annotation constraints with the grouping algorithms is a challenging and novel contribution.
> >
> > **References**
> >
> > [1]. Qi Huang, Hanze Zhang, Jiaqing Chen, and MJJBB He. Quantile regression models and their applications: A review. Journal of Biometrics & Biostatistics, 8(3):1–6, 2017.
> >
> > [2]. Roger Koenker and Gilbert Bassett Jr. Regression quantiles. Econometrica: journal of the Econometric Society, pp. 33–50, 1978.
> >
> > [3]. Shai Feldman, Stephen Bates, and Yaniv Romano. Calibrated multiple-output quantile regression with representation learning. Journal of Machine Learning Research, 24(24):1–48, 2023.
> >
> > [4]. Nicolai Meinshausen and Greg Ridgeway. Quantile regression forests. Journal of machine learning research, 7(6), 2006
> > [5]. Yandong Yang, Shufang Li, Wenqi Li, and Meijun Qu. Power load probability density forecasting using gaussian process quantile regression. Applied Energy, 213:499–509, 2018.
> >
> > [6]. Cao, Dong-Sheng, et al. Tree-based ensemble methods and their applications in analytical chemistry. TrAC Trends in Analytical Chemistry, 40: 158-167, 2012.
> >
> > [7]. Hong, Huixiao, et al. Prediction of estrogen receptor binding for 58,000 chemicals using an integrated system of a tree-based model with structural alerts. Environmental Health Perspectives, 110.1: 29-36, 2002.

---

### Review · Reviewer_CFgZ · 2025-07-31

**Summary Of Contributions:**

The paper proposes adapting active learning methods for material science purposes. The proposal consists of picking some samples from given groups, aiming for improved diversity of the samples picked and, as such, the quality of results, while reducing the annotation burden. Results are reported on several material synthesizing problems.

**Audience:**

No

**Broader Impact Concerns:**

There are no broader impacts addressed in the paper, but given the scope of it, I don’t think this is necessary. The work proposed is for a relatively narrow task of machine learning used for material science.

**Claims And Evidence:**

No

**Requested Changes:**

I feel the paper requires a major rewrite to improve its quality and clarity. The contributions in terms of machine learning are not clear at all. In its current state, I am unable to identify significant contributions of the paper, out of application proposed to material science, which is not clearly presented and is rather niche. Better motivations of the work, making it clear the potential value and impact of the solution proposed may make the paper more valuable. And all the notions of groups, which seem to be the original aspect of the proposal, should be better presented and all explanations should connect well with it.

**Strengths And Weaknesses:**

Strengths:
* Relatively straightforward practical use of active learning.
* The paper is correctly written (grammar, spelling and individual sentences).


Weaknesses:
* The paper is unclear and hard to follow. The motivations and contributions are not well presented. Although it may look good at a glance, as we start to dig into the topic, there are many missing and not very well explained concepts.
* More specifically, the formulation of the problem over the use of active learning over groups is not stated clearly. Standard active learning (Sec. 3.1) is stated well, but the explanation for groups (Sec. 3.3) is not clear and it is difficult to make sense of it. Maybe some rationale should be provided to clarify the operations presented. It seems to be some kind of stratified sampling that is proposed here, but I am not totally sure what we try to achieve.
* It is still not clear to me why MQR is used in the current context. The number of labels is limited in active learning, while using MQR requires a significant number of samples when the number of bins (quantiles) used is more than a few — enough samples are required to have a good estimate. MQR doesn’t seem to be a good design choice for the current setting, in my opinion. Relying on probabilistic approaches like Gaussian process regression, for example, would have provided a stronger way to estimate uncertainty reliably with a few samples.
* I don’t get the point of sections 4.1, 4.2 and 4.3, about active learning strategies. Explanations given in “Predictive Probability in Regression” part of 4.1 are rather trivial and common use of histograms for probability estimation. But for all the explanations of the “Prediction Intervals Probability” part, the purpose of it is not clear at all.
* The diversity contribution in Sec. 4.2 seems to be limited to the application of k-means++ to select N_batch centers from a set of samples, which is straightforward explained in one paragraph. I don’t get why we should re-explain with equations a rather classical clustering algorithm. This does not seem relevant here, as the method is used as is, with no particular modifications. However, this presentation is not clearly linked with the notion of groups that represents where lay the originality of the paper.
* Sec. 4.3 put it all together, but again the rationale is not clear. I find it rather hard to follow Algorithm 1 given the fact it refers to equations by numbers. I would have rather restated all equations directly in the Algorithm for self-contained and complete presentation of the approach. It would provide a clear overview and formal presentation of the approach. Steps 1 to 4 referred in the second paragraph of Sec. 4.3 are not given at all in Algorithm 1 or anywhere else. Things are not connecting well.
* The experimental setting and application are not well explained in Sec. 5, text provided lacks context. Although the real contribution seems the use of active learning for material science, we don’t have specific explanations on the formulation and meaning of this problem. Basically, for a reader not very familiar with this scientific problem (like myself), it gets impossible to figure out what’s going on with these results and whether they are significantly good or better than the baselines. Results presented in figures 3 to 6 show that the MQR-UD is converging faster than other approaches most of the time. But I am unable to interpret specifically if this is truly meaningful. And the fact that this application is very specific and that the proposed approach is not tested in other contexts makes its value quite narrow, application-wise.
* The technical contributions on the use of active learning are quite small for what I understand of the paper, as it is mostly reusing many existing notions in the current applicative context. There are no clear technical / algorithmic contributions to the paper, as far as I can see.

---

> ### Author Response · Authors · 2025-08-11
> **Response to Reviewer CFgZ**
>
> We sincerely appreciate the reviewer’s valuable suggestion. We will address the questions point by point.
>
> **Weaknesses**
> 1. We thank the reviewer for the valuable feedback. We understand that parts of the paper were not sufficiently clear in presenting the motivations and contributions. We have made a lot of revisions and will explain in the following questions.
>
> 2. Thanks for pointing this out. Based on our explanation in the Introduction, the synthesis process (a prior step of annotation) in these scientific scenarios is batch processing, which means that some samples in the same group are synthesised from the same experiment on the same device.  And we also added a set of comparison examples of this problem in the Introduction to clarify this. “The examples and comparison of the conventional AL framework and the constrained framework are shown in Figure 1. Assume we select the same number of samples under these two frameworks in one AL iteration, so that the total length of the blue blocks in the two sub-figures is the same. The length of the red dashed square represents the capability of each synthesizers. Since in the conventional AL framework, the selected sample count in some groups likely exceeds the capability of a single synthesizer (G1 in sub-figure (a)), and selected samples count in some other groups are much less than the capability of a synthesizer (G2-4 in sub-figure (a)), which leads running more synthesis process (number of red dashed square) and also waste of the synthesizer space each time. But in our constrained framework, we limit the sample selection to a fixed number of groups based on the count of the synthesisers, so that it is much more efficient. “
>
> We have added some explanation about our group constraints in the framework to make this clear in Section 3.3: “Our constrained AL framework is designed for scenarios where unlabelled data are grouped (e.g., synthesis batches). And only a limited number of groups can be selected per AL iteration due to experimental or resource constraints (e.g., number of available synthesizers). Accordingly, the AL selection process must address two levels of decision-making. (1). Which samples are most informative within each group? (2). Which groups to prioritise under a fixed group budget? The difference between the conventional AL framework and our constrained framework is that our selection space is structured by group instead of querying from the entire space of the unlabelled pool. Our selection is constrained to top-ranked groups based on aggregate informativeness. Specifically, in each AL iteration, the framework selects the top $N_\textrm{devices}$ groups and then queries top-scoring $N_\textrm{synthesis}$ samples from each group.”
>
> 3. The reason why we use MQR is that, compared to other uncertainty estimation methods, like Gaussian Process, MQR offers a flexible alternative for uncertainty estimation in regression that requires no distributional assumptions and exhibits robustness to outliers.  Even with limited labelled data in the early stage of AL, our experimental results demonstrate that effective sample selection can still be achieved. As AL iteration increases with more labelled data, MQR estimation becomes increasingly reliable. We have added the advantage of MQR to other uncertainty estimation methods/models in the Introduction. “Compared to other methods, MQR offers a flexible alternative for uncertainty estimation in regression that requires no distributional assumptions, exhibits robustness to outliers[1] and can be implemented using a wide range of regression models, including linear model[2], neural network[3], tree-based model[4], and Gaussian Process[5].”
>
> [contd.]

---

> > ### Author Response · Authors · 2025-08-11
> > **Response to Reviewer CFgZ**
> >
> > [contd.]
> >
> > 4. As we discussed in the Introduction, a lot of recent AL methods designed for classification that cannot be easily extended to regression tasks, the reason is basically due to the classifiers having the predictive probability for each class that can directly provide the model confidence for the AL acquisition function. For instance, some of the methods use this probability to generate pseudo labels [6][7] for unlabelled samples, or directly use this probability information to construct the uncertainty estimation, the simplest and popular ways like Margin[8], Entropy[9], Least Confident[10].
> >
> > The purpose of these sections 4.1, 4.2, 4.3 is to construct the interval predictive probability in regression so that we can directly apply a lot of AL strategies designed for classification to regression. In this case, the goal is to narrow the technical and application gaps between AL classification and regression. By using this simple and effective way of histograms for probability estimation through MQR, we could achieve this purpose. We think this is also one of our contributions in this paper, since our work can be further extended to many other different directions by using the interval probability with some advanced classification-designed AL to solve the regression problems. Our paper focuses on the constrained AL framework, but our AL method is also practical for the normal AL framework without constraints. Since in our scenarios, the different data groups might have a similar distribution in output space, but also may have relatively discrete and separate distributions. We thus use the minimum and maximum prediction values of each group to construct the prediction interval. But in the future, if we don’t use the data that have clearly different groups, we could use the minimum and maximum prediction values of all samples to construct intervals and probability and make this method a general AL strategy.
> >
> > To clarify this, we have added our motivation for interval predictive probability in Section 4.1. “To adapt these uncertainty estimation methods from classification to regression, we employ a simple yet effective density-based probability approach using the multiple outputs of MQR to capture posterior probabilities in prediction intervals for AL regression.  ” and in the Discussion, we add “Although we developed our MQR-UD method within the constrained AL framework, this AL strategy can also be applied to general AL frameworks without such constraints. We use the minimum and maximum MQR predictions within each group to construct the prediction interval, as different data groups may share similar output distributions or exhibit relatively distinct and separate ones. But in the future, if we don’t use the data that have clearly different groups, we could use the minimum and maximum prediction values of all samples to construct intervals and probability and make this method a general AL strategy.”
> >
> > 5. Thanks for pointing this out. The equations here are not a classical clustering algorithm, since we did not do the clustering iterations. We firstly use the k-means++ to identify the cluster centres and then assign each sample to the closest centre, and we do not have the actual clustering iterations in our algorithm, considering the computational complexity in AL. Thus, here the equations are for better reproducibility and to avoid misleading that we did the clustering algorithm.
> >
> > Section 4.2 introduces the diversity component in its general form before applying it to the group-constrained setting, which mirrors the structure of our AL framework description in Section 3. We first define the general AL formulation and then adapt it into our constrained framework. The general form of diversity contribution serves as a modular block, which is later combined with the uncertainty component under the group constraints defined in Section 4.3. This separation was intended to make the framework self-contained and to clarify that our method is built from the standard AL framework with explicit adaptations for grouped data. In our original submission, we have mentioned that “We first define the diversity contribution generally (in terms of a general unlabelled data set $\mathbb{U}$ and query batch size $N_{\textrm{batch}}$), which will be combined with the uncertainty component and the constrained AL framework in the following section. “  We also added “Within our constrained AL framework, the diversity component is applied separately to each group, defined in Section 4.3.” at the end of section 4.2. The link of diversity with groups is defined in Section 4.3 with equations 27 and 28, which are adapted from equations 25 and 26 in Section 4.2.
> >
> > [contd.]

---

> > > ### Author Response · Authors · 2025-08-11
> > > **Response to Reviewer CFgZ**
> > >
> > > [contd.]
> > >
> > > 6. We removed the sentence “These were defined generally so that, for example, the acquisition function could be applied with any of the baseline approach acquisition scores in Section 3.2. ” in Section 4.3 that might cause the misleading.  We added detailed descriptions in every line of Algorithm 1 to improve understanding and clarify the connections. Also, we have added Steps 1 to 4 in the Algorithm to represent our constrained framework better.
> > >
> > > 7. We have made significant changes to clarify the experimental setting, including adding a figure (now figure 1) and modifying the original figure 1 (now figure 2). We added the meaning of this problem at the beginning of Section 5: “Our constrained AL framework limits the selection to $N_{\textrm{devices}}$ groups, where $N_{\textrm{devices}}$ represents the number of available synthesizers that could generate physical samples. Since in each group we select $N_{\textrm{synthesis}}$ samples matching the capacity of the synthesizers, our constrained framework can help improve synthesizer utilisation. In scientific discovery tasks, when exploring a larger sample space, the number of groups is usually larger than the number of synthesizers. Thus, applying our constrained AL framework is very valuable since it could save lab resources and improve efficiency in real applications. When comparing baselines within the same constrained AL framework, if the AL method has faster convergence of the model performance, it indicates that it can achieve the same model performance as other AL methods with fewer labelled data, resulting in lower synthesis and annotation costs in the task.”
> > >
> > > We also added the setting in Section 5. “We tested $N_\textrm{devices} = 1$ and $5$ in our experiments to represent single and multiple synthesizers scenarios. Since Superconductivity-S has smaller group sizes than other datasets, we use $N_\textrm{synthesis} = 5$ for Superconductivity-S and $N_\textrm{synthesis} = 10$ for other datasets.” The data setting for splitting datasets into groups is described in Section 5.1.
> > >
> > > Our framework is also meaningful to other domains. In real-world applications, annotation is often conducted in groups across many scenarios. For instance, medical data can be grouped by patient demographics (e.g., age), illness type, or medications, with different clinician groups assigned to their respective specialities. Similarly, in crowdsourcing, tasks are divided into blocks based on annotators' backgrounds and expertise. Our constrained AL framework leverages these natural groupings to improve annotation efficiency by assigning a more balanced number of tasks to different annotators.“  This explanation has been added to the Discussion.
> > >
> > > 8. Thanks for pointing this out. We appreciate the chance to clarify our contributions. We respectfully disagree that there are no technical or algorithmic contributions. Although we employ components from existing methods (e.g., k-means++, MQR), the novelty of our work lies in (i) formalising the constrained group-wise synthesis setting, which can be widely applied to different scientific scenarios. (ii) Designed the novel AL acquisition function based on MQR. (iii) Applied the classification-designed AL strategy (entropy) to regression, which is a crucial attempt since the interval predictive probability in our study can be applied to the other classification-designed AL, which can narrow the gap between AL in regression and classification. To the best of our knowledge, this is the first work to address AL under such constraints in scientific tasks.
> > >
> > > **Requested Changes**
> > > 1. Thanks for pointing out that our motivation is not clear. We have rewritten our Introduction and added extensive explanations. For example, we added: “Such constraints are often defined by experimental design and known in advance. The input features for the ML model, such as composition, structure, or synthesis conditions, can typically be obtained prior to synthesis from designed experimental parameters, reference databases, or computational simulations. In contrast, the annotation requires synthesising the physical samples under these constraints first, followed by experimental characterisation to get labels. Typical batch AL has the potential to accommodate such constrained settings by grouping query tasks.”  This clearly explained the real setting of the synthesis.  We also added Figure 1: Comparison of sample selection in one AL iteration between the conventional AL framework and the constrained AL framework, to better introduce our problem.  We refined our Figure 2 (The constrained AL framework).
> > > We also added some of the points in our contribution part:
> > >
> > > [contd.]

---

> > > > ### Author Response · Authors · 2025-08-11
> > > > **Response to Reviewer CFgZ**
> > > >
> > > > [contd.]
> > > >
> > > > We also added some of the points in our contribution part:
> > > > Our contributions are:
> > > > 1). We introduce a framework and a simple AL strategy to accelerate data generation with constraints by limiting sample selection to predefined groups, which guarantees experimentation efficiency under experimental equipment or resource limits. The other AL methods can also be applied to our constrained framework.
> > > > (2). We leverage MQR to get the probability of the prediction intervals for regression tasks. This is similar to the classifier's predictive probability, which is used as significant information in AL. Therefore, it bridges the gap between AL regression and classification methods and allows us to extend some classification-designed AL methods for regression.
> > > > (3). Experiments demonstrate its effectiveness for scientific tasks. We also provide an analysis of scenarios where our method may not be applicable. In addition, it can also be adapted to other applications where the data processing or annotation is group-wise.
> > > >
> > > >
> > > > **References**
> > > >
> > > > [1]. Qi Huang, Hanze Zhang, Jiaqing Chen, and MJJBB He. Quantile regression models and their applications: A review. Journal of Biometrics & Biostatistics, 8(3):1–6, 2017.
> > > >
> > > > [2]. Roger Koenker and Gilbert Bassett Jr. Regression quantiles. Econometrica: journal of the Econometric Society, pp. 33–50, 1978.
> > > >
> > > > [3]. Shai Feldman, Stephen Bates, and Yaniv Romano. Calibrated multiple-output quantile regression with representation learning. Journal of Machine Learning Research, 24(24):1–48, 2023.
> > > >
> > > > [4]. Nicolai Meinshausen and Greg Ridgeway. Quantile regression forests. Journal of machine learning research, 7(6), 2006.
> > > >
> > > > [5]. Yandong Yang, Shufang Li, Wenqi Li, and Meijun Qu. Power load probability density forecasting using gaussian process quantile regression. Applied Energy, 213:499–509, 2018.
> > > >
> > > > [6]. Wang, Tianyang, et al. Boosting active learning via improving test performance. Proceedings of the AAAI Conference on Artificial Intelligence, Vol. 36. No. 8, 2022.
> > > >
> > > > [7]. Liu, Zhuoming, et al. Influence selection for active learning. Proceedings of the IEEE/CVF international conference on computer vision, 2021.
> > > >
> > > > [8]. Campbell, Colin, Nello Cristianini, and Alex Smola. Query learning with large margin classifiers. ICML, Vol. 20. No. 0., 2000.
> > > >
> > > > [9]. C.E. Shannon. A mathematical theory of communication. Bell System Technical Journal, 27:379–423,623–656, 1948.
> > > >
> > > > [10]. Freeman, Linton G. Elementary applied statistics, 1965.

---

### Author Response · Authors · 2025-08-11
**The main changes in the new version of our manuscript**

We summarise our changes here:

**Main Changes**

1. Added Figure 1 in the Introduction.

2. Refined Figure 2 in the Introduction.

3. Added more motivation in the Introduction, and some other methods in the Related Work.

4. In Sections 3.3 and 4.3, we added some insights into designing our framework and algorithm.

5. Added more text descriptions in Algorithm 1 and several points in the Discussion.

6. Added Appendix C.4 about time and memory usage scaling.

**Other Changes**

1. We removed “or a fixed capacity to experimental equipment [1]” in the Introduction.

2. We removed the inappropriate citation [2] in the Introduction.

3. We found it might be confusing that synthesis and annotation are mixed together. The constraints are from the synthesis, which is a prior step of the annotation. Thus, we consider that the synthesis constraints a part of the process of getting labels. To avoid confusion, we changed all $N_\textrm{oracles}$ to $N_\textrm{devices}$ in the paper, since oracles always refer to the annotation process. But in this context, it is not very accurate.

**References**

[1] Anna Papiez, Michal Marczyk, Joanna Polanska, and Andrzej Polanski. Batchi: Batch effect identification in high-throughput screening data using a dynamic programming algorithm. Bioinformatics, 35(11):1885–1892, 2019.

[2] Michael J Geuenich, Dae-won Gong, and Kieran R Campbell. The impacts of active and self-supervised learning on efficient annotation of single-cell expression data. Nature Communications, 15(1):1014, 2024.

---

> ### Author Response · Authors · 2025-08-14
> **The changes in the second revision of the manuscript**
>
> Changes:
>
> 1. Added several citations in the third paragraph of the Introduction.
> 2. Added and changed citations in the Related Work.
> 3. In related Work: Move the description of RT-AL series methods from the subtitle "Hybrid AL strategy" to "Other Criteria in AL".
> 4. Updated some wording.
> 5. All changes are highlighted in red text.

---

### Decision · Action_Editor_kMv2 · 2025-10-24

**Recommendation:** Reject

**Audience:**

Yes

**Audience Explanation:**

This paper may suits a more application-domain journal better. But still very few, but some, of TMLR's audience may be interested in the paper.

**Claims And Evidence:**

No

**Claims Explanation:**

While the paper presents a clearly motivated application and provides empirical evidence supporting its effectiveness, the central claim that the proposed approach can “accelerate annotation with data generation constraints” is only partially supported by convincing and clear evidence. The experiments, though positive, are confined to a narrow, domain-specific setting, and the method’s design remains largely heuristic---repurposing established techniques such as quantile regression for uncertainty estimation and k-means++ for diversity selection-----without providing strong theoretical justification or broader empirical validation. As a result, the work’s general applicability "accelerate annotation with data generation constraints" remains uncertain.